# Engineered ACE2 receptor therapy overcomes mutational escape of SARS-CoV-2

Yusuke Higuchi[1,10], Tatsuya Suzuki[2,10], Takao Arimori [3,10], Nariko Ikemura[1,10], Emiko Mihara [3], Yuhei Kirita [4], Eriko Ohgitani[5], Osam Mazda [5], Daisuke Motooka[6], Shota Nakamura[6], Yusuke Sakai [7], Yumi Itoh[2], Fuminori Sugihara[8], Yoshiharu Matsuura [9], Satoaki Matoba [1], Toru Okamoto [2✉], Junichi Takagi [3✉] & Atsushi Hoshino [1✉]

SARS-CoV-2 has mutated during the global pandemic leading to viral adaptation to medications and vaccinations. Here we describe an engineered human virus receptor, ACE2, by mutagenesis and screening for binding to the receptor binding domain (RBD). Three cycles of random mutagenesis and cell sorting achieved sub-nanomolar affinity to RBD. Our structural data show that the enhanced affinity comes from better hydrophobic packing and hydrogen-bonding geometry at the interface. Additional disulfide mutations caused the fixing of a closed ACE2 conformation to avoid off-target effects of protease activity, and also improved structural stability. Our engineered ACE2 neutralized SARS-CoV-2 at a 100-fold lower concentration than wild type; we also report that no escape mutants emerged in the co-incubation after 15 passages. Therapeutic administration of engineered ACE2 protected hamsters from SARS-CoV-2 infection, decreased lung virus titers and pathology. Our results provide evidence of a therapeutic potential of engineered ACE2.

---

[1] Department of Cardiovascular Medicine, Graduate School of Medical Science, Kyoto Prefectural University of Medicine, Kyoto, Japan. [2] Institute for Advanced Co-Creation Studies, Research Institute for Microbial Diseases, Osaka University, Osaka, Japan. [3] Laboratory for Protein Synthesis and Expression, Institute for Protein Research, Osaka University, Osaka, Japan. [4] Department of Nephrology, Graduate School of Medical Science, Kyoto Prefectural University of Medicine, Kyoto, Japan. [5] Department of Immunology, Graduate School of Medical Science, Kyoto Prefectural University of Medicine, Kyoto, Japan. [6] Department of Infection Metagenomics, Research Institute for Microbial Diseases, Osaka University, Osaka, Japan. [7] Department of Veterinary Pathology, Yamaguchi University, Yamaguchi, Japan. [8] The Core Instrumentation Facility, Research Institute for Microbial Diseases, Osaka University, Osaka, Japan. [9] Department of Molecular Virology, Research Institute for Microbial Diseases, Osaka University, Osaka, Japan. [10] These authors contributed equally: Yusuke Higuchi, Tatsuya Suzuki, Takao Arimori, Nariko Ikemura. ✉email: toru@biken.osaka-u.ac.jp; takagi@protein.osaka-u.ac.jp; a-hoshi@koto.kpu-m.ac.jp

Coronavirus disease 2019 (COVID-19) has spread across the world as a tremendous pandemic and presented an unprecedented challenge to human society. The causative agent of COVID-19, SARS-CoV-2 is a single-stranded positive-strand RNA virus that belongs to lineage B, clade 1 of the beta-coronavirus genus[1–3]. The virus binds to host cells through its trimeric spike glycoprotein composed of two subunits; S1 is responsible for receptor binding and S2 for membrane fusion[4]. Angiotensin-converting enzyme 2 (ACE2) is lineage B clade 1 specific receptor including SARS-CoV-2[3]. The receptor binding domain (RBD) of S1 subunit directly binds ACE2, therefore, it is the most important targeting site to inhibit viral infection. In fact, the RBD is the common binding site of effective neutralizing antibodies identified from convalescent patients[5–7]. Among RNA viruses, coronaviruses uniquely encode viral RNA proofreading exoribonuclease that may have benefit to large ~30 kb genome[8]. However, SARS-CoV-2 still have high mutation rates ~$1 \times 10^{-3}$ substitutions per site per year[9], which are correlated with high evolvability including the acquisition of anti-viral drug resistance. Mutations in the spike gene can lead to the SARS-CoV-2 adaptation to neutralizing antibodies even in the cocktail treatment[10] and polyclonal convalescent plasma[11–13]. Similar to anti-receptor binding domain (RBD) antibodies, extracellular domain of ACE2, soluble ACE2 (sACE2), can also be used to neutralize SARS-CoV-2 as a decoy receptor. The therapeutic potency was confirmed using human organoid[14], and now Apeiron Biologics conducts European phase II clinical trial of recombinant sACE2 against COVID-19[15]. Most importantly, sACE2 has a great advantage over antibodies due to the resistance to virus escape mutation. The mutant escaping from sACE2 should have limited binding affinity to cell surface native ACE2 receptors, leading to a diminished or eliminated infectivity. Unfortunately, many reports, including our current study, have revealed that the binding affinity of wild-type (WT) sACE2 to the SARS-CoV-2 spike RBD is much weaker ($K_D$ ~20 nM) than that of clinical grade antibodies[4,16–18]. Thus, the therapeutic potential of the WT sACE2 as a neutralizing agent against SARS-CoV-2 is uncertain.

In this work, we perform directed evolution for ACE2 in 293T cells and achieve ~100-fold higher binding affinity for the RBD. Engineered ACE2 based on high affinity ACE2 mutant and human IgG1-Fc effectively neutralizes SARS-CoV-2 and exhibits the therapeutic effect in a COVID-19 model hamster. It also prevents the emergence of escape mutation and effectively neutralizes current N501Y and E484K-based mutants as well as escape mutants from COVID-19 convalescent plasma. Engineering ACE2 decoy receptors with human cell-based directed evolution is a promising approach to develop a SARS-CoV-2 neutralizing drug that has affinity comparable to monoclonal antibodies yet displaying resistance to escape mutations of virus.

## Results

**Directed evolution of ACE2 for higher affinity to the RBD.** Here we conducted human cell-based directed evolution to improve the binding affinity of ACE2 to the spike RBD with the combination of surface display of mutagenized library and fluorescence-activated cell sorting (FACS). Among various host cells, the budding yeast is popular in directed evolution due to the efficient surface display and greater molecular diversity[19]. However, posttranslational modifications such as glycosylation are substantially different between yeast and mammalian cells, which could alter the protein activity including the binding affinity[20]. In consideration of future drug development in mammalian cells, we developed the screening system based on 293T cells. The protease domain (PD) of ACE2 is known to harbor the interface to viral spike protein, located in the top-middle part of ACE2

ectodomain. In this study, ACE2 residues 18–102 and 272–409, referred to as PD1 and PD2, respectively, were mutagenized independently. Synthetic signal sequence and HA tag were appended and restriction sites were introduced in both sides of PD1 and PD2 by optimizing codon (Supplementary Fig. 1a). We used error-prone PCR to mutagenize the protease domain of ACE2 at the rate of ~10 mutations per 1 kb, then inserted the fragment into the introduced restriction site. The reaction sample was transformed to competent cell, generating a library of ~$10^5$ mutants. Mutant plasmid library was packaged into lentivirus, followed by expression in human 293T cells in less than 0.3 MOI (multiplicity of infection) to yield no more than one mutant ACE2 per cell. Cells were incubated with recombinant RBD of SARS-CoV-2 spike protein fused to superfolder GFP (sfGFP; Fig. 1a). We confirmed the level of bound RBD-sfGFP and surface expression levels of HA-tagged ACE2 with Alexa Fluor 647 in two-dimensional display of flow cytometry. Top 0.05% cells showing higher binding relative to expression level were harvested from ~$5 \times 10^7$ cells by FACS. To exclude the structurally unstable mutants, cells with preserved signal of surface ACE2 were gated. Genomic DNA was extracted from collected cells and mutagenized again to proceed to the next cycle of screening (Fig. 1b).

Random mutagenesis screening for PD1 was performed 3 times and mutated sequences from top 0.05% population were reconstructed into the backbone plasmid and expressed in 293T cells individually. The binding capacity to the RBD-sfGFP was verified in 100-300 clones. As the selection cycle advances, the two-dimensional distribution of library cells in flow cytometry became broader and higher in RBD-binding signal, and individual clone validation identified several mutants with higher binding capacity (Supplementary Fig. 1b). To estimate potential neutralization activity in the form of sACE2, we first generated fusion protein of the soluble extracellular domain of mutagenized ACE2 residues 18-615 and sfGFP (sACE2-sfGFP) and used them to compete with the cell-surface WT ACE2 for the RBD binding. To this end, the concentration of each mutant sACE2-sfGFP in the cultured medium from transfected cells was quantitatively standardized with sfGFP signal, serially diluted, preincubated with RBD-sfGFP for 30 min, and then transferred to WT ACE2 expressing 293T cells. After 30 min, the RBD-bound cells were analyzed in flow cytometry. We evaluated 5 independent clones including one from the first selection (1–19), two from the second selection (2–51 and 2–53), and two from the third selection (3J38 and 3N39). As shown in Supplementary Fig. 1c and Supplementary Table 1, all mutant sACE2s showed higher competing activity toward the cell binding of RBD-sfGFP than the WT sACE2. Furthermore, there was a progressive increase in the activity as the selection cycle advanced, with a clone 3N39 in the 3rd selection showing the highest activity. In a separate experiment, we conducted second mutagenesis on the top hit clone in the first screening, 1–19, followed by the evaluation of the RBD binding, but the distribution of the resultant library cells showed little change, and we could not isolate clones showing higher binding signals than the bulk of top 0.05% (Supplementary Fig. 1d). We also performed PD2 mutagenesis on both the bulk of top 0.05% and the clone 3N39, hoping to find tighter binders. Again, the binding distribution of the PD2 library cells was similar to the basal cells (Supplementary Fig. 1b). A recent study reported, via deep mutational scanning, that several specific mutations in PD2 were enriched in high RBD-binding clones[17]. Even when we applied these mutations in 3N39, it failed to further improve the capacity of the RBD neutralization (Supplementary Fig. 1e), suggesting that the search for the high affinity mutants with the mutated PD1 was already exhaustive after three cycles and isolation of clones with further increased

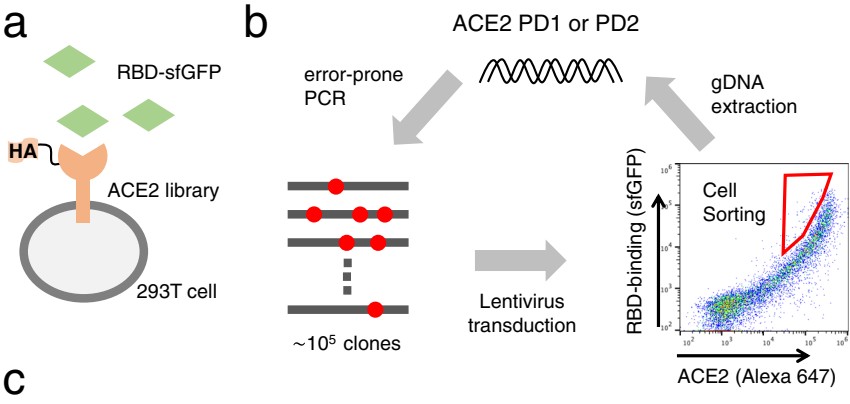

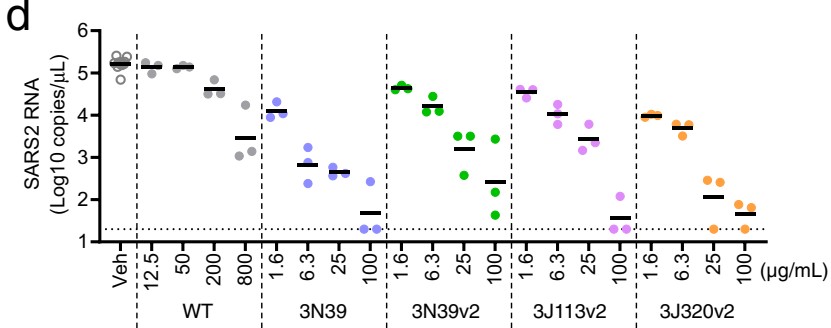

| variant | mutations | $K_D$ (nM) | IC50 (µg/ml), Psudovirus | |
|---|---|---|---|---|
| | | | SARS-CoV-2 | SARS-CoV-1 |
| wild-type | | 17.63 | 24.8 | 6.8 |
| 3N39 | A25V, K26E, K31N, E35K, N64I, L79F, N90H | 0.29 | 0.056 | 0.011 |
| 3N39v2 | A25V, K31N, E35K, L79F | 0.64 | 0.082 | 0.027 |
| 3J113v2 | K31M, E35K, Q60R, L79F | 1.14 | 0.33 | 5.3 |
| 3J320v2 | T20I, H34A, T92Q, Q101H | 3.98 | 0.068 | 0.022 |

**Fig. 1 Directed evolution to generate high affinity ACE2 in 293T cells. a** ACE2 mutant library was expressed in 293T cells and incubated with the RBD of SARS-CoV-2 fused to superfolder GFP (sfGFP). **b** Error-prone PCR amplification of ACE2 protease domain induced random mutations. Mutant library-transduced cells were incubated with the RBD-sfGFP. Top 0.05% population with high level of bound RBD-sfGFP was sorted and underwent DNA extraction, followed by next cycle mutagenesis. Cell sorting was conducted by gating on forward scatter (FSC)-H and FSC-A to exclude doublets, followed by gating on Alexa 647 for HA-ACE2 expression and sfGFP for RBD-binding. **c** The value of $K_D$ and IC50 against pseudovirus of SARS-CoV-2 and SARS-CoV-1. **d** Neutralization potency to authentic SARS-CoV-2 was analyzed in Vero6E/TMPRSS2 cells. Data are mean of $n = 3$ technical replicates.

affinity toward RBD may not be possible with the current strategy.

**High affinity ACE2 and its neutralization of SARS-CoV-2.** In the end, we identified 3 highest binding mutants, 3N39, 3J113 and 3J320 in the top population from the 3rd library (Supplementary Fig. 2a). To identify essential mutations, each mutation was altered back to WT in these 3 mutants individually or in combination and the resultant partially back-mutated ACE2 proteins were compared with the original mutant in the RBD-competing assay (Supplementary Fig. 2b–g). These experiments lead to the identification of essential mutations for each original mutant, and the mutants carrying only the essential mutations were referred to as v2 hereafter. Supplementary Table 2 shows the list of mutated residues present in three final candidates (3N39v2, 3J113v2, and 3J320v2) along with some of the high-affinity ACE2 mutants reported previously by other groups[17, 18]. Next, we characterized their binding affinities by surface plasmon resonance (SPR). The $K_D$ value of WT sACE2 was 17.63 nM, whereas those of mutants were determined to be from 0.29 nM to 3.98 nM (Fig. 1c,

Supplementary Fig. 3), confirming that the increased RBD-binding activity of these mutants were in fact due to the increased binding affinity up to ~100-fold. The SPR sensorgrams indicates that high affinity mutants all have very slow off-rate compared to the WT ACE2. It is commonly believed that ACE2 binding can occur when RBD is in a "up" conformation in the spike trimer, and most of the cryo-EM images of the ACE2-spike complex have only one ACE2 bound per spike trimer[21]. In fact, a mass photometry analysis of WT ACE2-spike trimer complex reveals that it exists as the mixture of complexes with varying ratios, with 1:1 complex as the major species (Supplementary Fig. 4, second panel from the top). Recently, Guo et al showed that the spike can assume "three-up" conformation with all RBDs occupied by ACE2[22]. However, this architecture was possible because the ACE2 was trimerized by an artificial C-terminal fusion with a foldon domain before the encounter with the spike protein. Therefore, it remained unclear if all the spike RBDs on the surface of a virus can be occupied by monomeric ACE2 proteins. When we performed the same mass photometry analysis on the spike trimer incubated with either 3N39 or 3N39v2 mutants at only 20% excess amount, both of them behaved predominantly as

**Table 1 Data collection and refinement statistics.**

| | 3N39 ACE2-RBD complex (PDB: 7dmu) |
|---|---|
| **Data collection**[a] | |
| Space group | $P4_3$ |
| Cell dimensions | |
| $a, b, c$ (Å) | 227.8, 227.8, 147.0 |
| Resolution (Å) | 48.13–3.20 (3.28–3.20)[b] |
| $R_{merge}$ | 0.29 (7.03) |
| $I / \sigma I$ | 9.38 (0.56) |
| CC1/2 (%) | 99.9 (29.3) |
| Completeness (%) | 99.9 (99.8) |
| Redundancy | 32.5 (28.9) |
| **Refinement** | |
| Resolution (Å) | 47.91–3.20 |
| No. reflections | 123,205 |
| $R_{work} / R_{free}$ (%) | 17.9/19.8 |
| No. atoms | |
| Protein | 12,855 |
| Carbohydrates | 437 |
| Others | 14 |
| $B$-factors | |
| Proteins | 132.9 |
| Carbohydrates | 209.0 |
| Others | 152.5 |
| R.m.s. deviations | |
| Bond lengths (Å) | 0.007 |
| Bond angles (°) | 0.960 |

[a]Four datasets collected from a single crystal were merged.
[b]Values in parentheses are statistics of the highest-resolution shell.

3:1 stoichiometric complex (Supplementary Fig. 4, the bottom two panels). This result clearly indicates that all spike RBDs on the SARS-CoV-2 can be saturated by stoichiometric amount of ACE2 protein when the affinity is high enough, highlighting the potential of the mutant ACE2 as the soluble virus-neutralizing agent.

Having a few high affinity ACE2 mutants in hand, we next decided to assess their potentials as anti-SARS-CoV-2 therapeutic agents. First, we confirmed that none of the mutants were structurally unstable compared to WT ACE2 by measuring the $T_m$ values in a thermal denaturation experiment (Supplementary Fig. 5). In fact, mutants 3J113v2 and 3J320v2 showed $T_m$ values higher than the WT. Next we formulated our high affinity mutant sACE2s as human IgG1 Fc fusion (ACE2-Fc), as recombinant soluble ACE2 (rsACE2) monomer was reported to have a fast clearance rate in human blood with a half-life of hours[23, 24]. We then evaluated their efficacy in neutralizing SARS-CoV-2 infections. Mutant ACE2-Fcs directly inhibited the binding of full-length spike trimer to cell-surface ACE2 with more than 100-fold increase in the blocking efficacy from the WT (Supplementary Fig. 6). Pseudotyped SARS-CoV-2 neutralization assay in ACE2-expressing 293T cells exhibited that $IC_{50}$ values of WT, 3N39, 3N39v2, 3J113v2 and 3J320v2 were 24.8, 0.056, 0.082, 0.33 and 0.068 µg/ml, respectively (Fig. 1c, Supplementary Fig. 7a). All mutants except for 3J113v2 were also capable of strongly inhibiting the infection of pseudotyped SARS-CoV-1, another ACE2-dependent coronavirus, indicating wide-range of therapeutic potential of 3N39v2 and 3J320v2 mutants (Fig. 1c, Supplementary Fig. 7b). When the neutralization potential against the authentic SARS-CoV-2 in TMPRSS2-expressing VeroE6 cells was evaluated, each mutant ACE2-Fc demonstrated significant neutralizing effect even in 100-fold lower concentration than WT (Fig. 1d).

**Crystal structure of mutant ACE2 and RBD complex.** In order to know the structural basis for the affinity enhancement, we next determined the structure of the highest affinity ACE2 mutant,

3N39, in complex with the SARS-CoV-2 RBD. The crystal structure of the complex was solved at 3.2-Å resolution (Fig. 2a, Table 1). Two 3N39-RBD complexes are contained in the asymmetric unit and they are structurally indistinguishable with a root mean square deviation (RMSD) value of 0.176 Å for 672 Cα atoms. The crystal structure confirms that 3N39 binds to RBD using the same interface as WT ACE2 reported previously[16, 25–27]. Among the 7 mutated residues in 3N39, the side chains of E26, I64, and H90 are exposed to solvent and not involved in either inter- or intra-molecular interactions, corroborating their non-essential nature in the affinity enhancement. In contrast, mutation sites K31N and E35K are located at the center of the binding interface. In the WT structure, E35 forms an intramolecular salt bridge with the K31 in the preceding helical turn (Fig. 2b, right), making it less favorable for forming direct intermolecular hydrogen bonding with Q493 in the RBD. This salt bridge is lost by the simultaneous mutations of K31N and E35K in 3N39, and K35 now forms a direct inter-molecular hydrogen bond exclusively with Q493 of RBD (Fig. 2b, left). This results in small but significant ~1 Å approach of the α1 helix of ACE2 toward RBD, likely contributing to the affinity enhancement of the mutant. As for L79F and A25V, both mutations are conservative in nature but accompany acquisition of larger sidechains. In WT ACE2, L79 forms a small hydrophobic pocket with Y83 and M82 that accommodates F486 of RBD, which has been suggested to be a key determinant of SARS-CoV-2 that gives higher affinity than SARS-CoV-1 (Fig. 2c, right)[25]. This hydrophobic contact became even more extensive due to the L79F mutation in 3N39. In addition, A25V mutation fills a space at the back of the pocket together with L97 (Fig. 2c, left). It is thus assumed that these mutations collectively lead to the 100-fold increase in the overall affinity, which is equivalent to 2.76 kcal/mol.

Among 4 ACE2 mutants that gained enhanced binding toward SARS-CoV-2 RBD, 3J113v2 was unique in that its binding toward RBD of SARS-CoV-1 remained the same as WT (Supplementary Fig. 7b). Structurally, K31M/E35K in 3J113 would act similarly to the K31N/E35K in 3N39, freeing the intramolecular salt bridge and reorienting the K35 toward Q493 as described above. However, the larger M31 sidechain may not be favored when binding to SARS-CoV-1 RBD, where the abutting L455 is mutated to much larger Y442, while small N31 in 3N39 can be accommodated and may even make favorable hydrogen bond with Y442 (Supplementary Fig. 8).

In the crystal structure, 3N39 adopts a "closed" conformation where the enzymatic active site is completely inaccessible due to the closure of the substrate binding groove (Supplementary Fig. 9a, left). This is very different from the "open" structures of WT ACE2 found in both RBD-bound (e.g., 6m0j) and unbound (e.g., 1r42) states (Supplementary Fig. 9a, right). Interestingly, the same closed structure was reported with WT ACE2 bound by an inhibitor MLN-4760 (Supplementary Fig. 9a, center)[28]. In fact, superposition of the two structures at the protease domain resulted in a surprisingly low RMSD value of 0.471 Å for 540 Cα atoms (Supplementary Fig. 9b). In the 3N39 ACE2 structure, the active site Zn is intact but there was no density corresponding to any substrate-like compounds in the pocket except for a sulfate ion (Supplementary Fig. 9c). This observation made us to wonder if the closed conformation of the 3N39 mutant contributed to its increased RBD binding ability, although the active site groove was far away from the RBD-binding interface (Supplementary Fig. 9a). In order to test this idea, we designed an ACE2 mutant that is fixed in the closed conformation by introducing S128C/V343C double mutations in WT ACE2 (Fig. 2d). Successful formation of the designed disulfide bond was inferred by an upward band shift of the mutant in a non-reducing SDS-PAGE (Supplementary Fig. 9d), as well as its complete loss of ACE2 activity

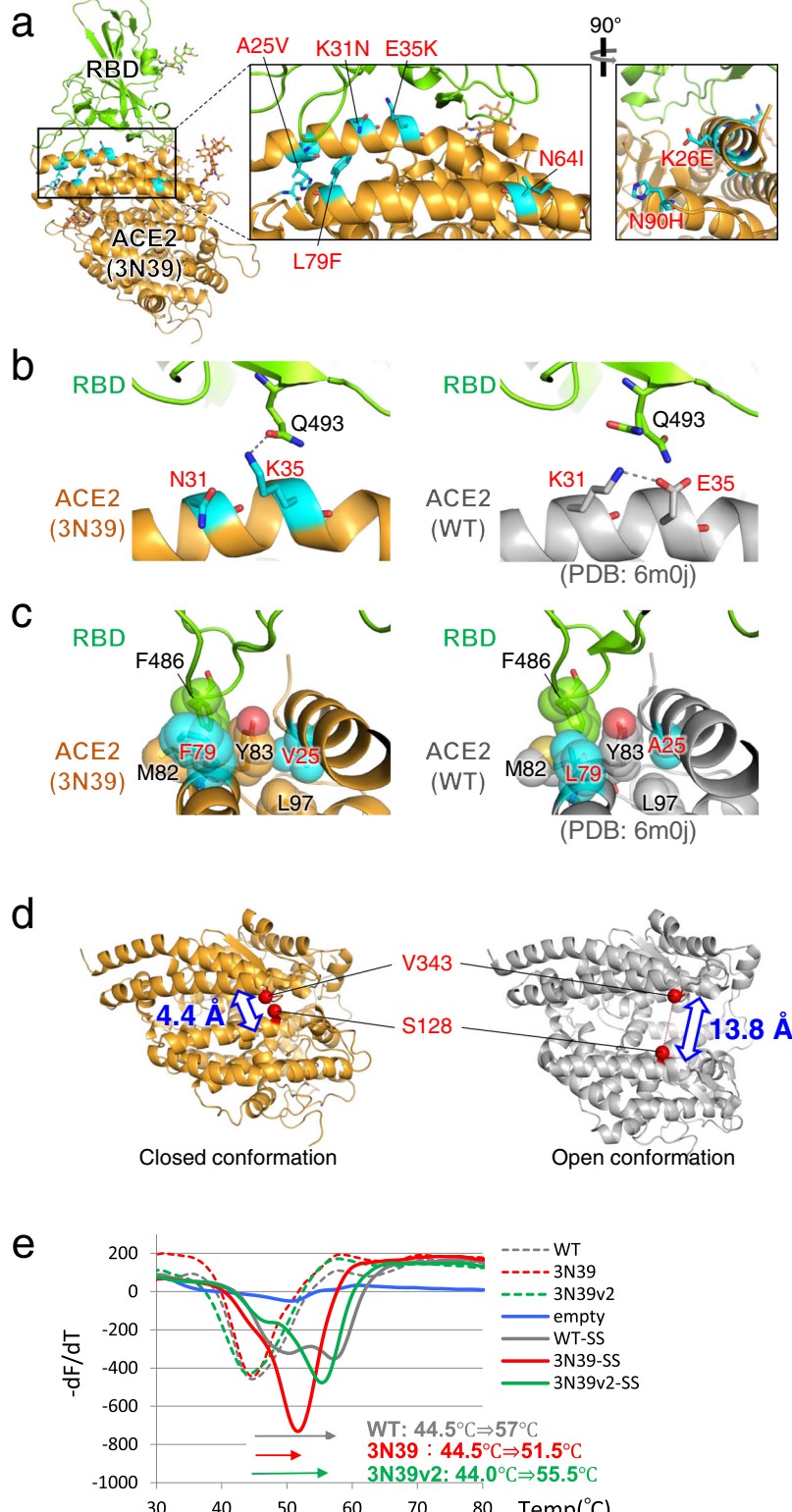

**Fig. 2 Structural analysis of 3N39 mutant in complex with RBD. a** Overall structure. 3N39 ACE2 and RBD are shown in orange and green, respectively. The mutated residues in 3N39 are shown as cyan stick models. The expanded views of the PD1 region are provided in the inset. **b** Structural comparison of the K31N/E35K mutation site in 3N39 (left panel) with its corresponding site in WT (right panel). Hydrogen-bonding interactions (within 3.0 Å) are indicated by dashed lines. **c** Structural comparison of the L79F/A25V mutation site in 3N39 (left panel) with its corresponding site in WT (right panel). F486 residue of RBD and hydrophobic residues composing the F486 binding pocket of ACE2 are shown as stick models with transparent sphere models. **d** Comparison of the distances between Cβ atoms of S128 and V343 residues in the closed and open conformations. **e** Stabilizing effect of S138C/V343C mutation. Various versions of ACE2-His proteins, with (solid lines) or without (dotted lines) the S138C/V343C disulfide mutation (SS), were subjected to the differential scanning fluorimetry using SYPRO™ Orange as the probe dye. The Tm value for each mutant is estimated by the peak temperature of the -dF/dT plot, and the Tm shift caused by the SS mutation is shown at the bottom. The experiments were independently performed three times and similar results were obtained. One representative data were shown.

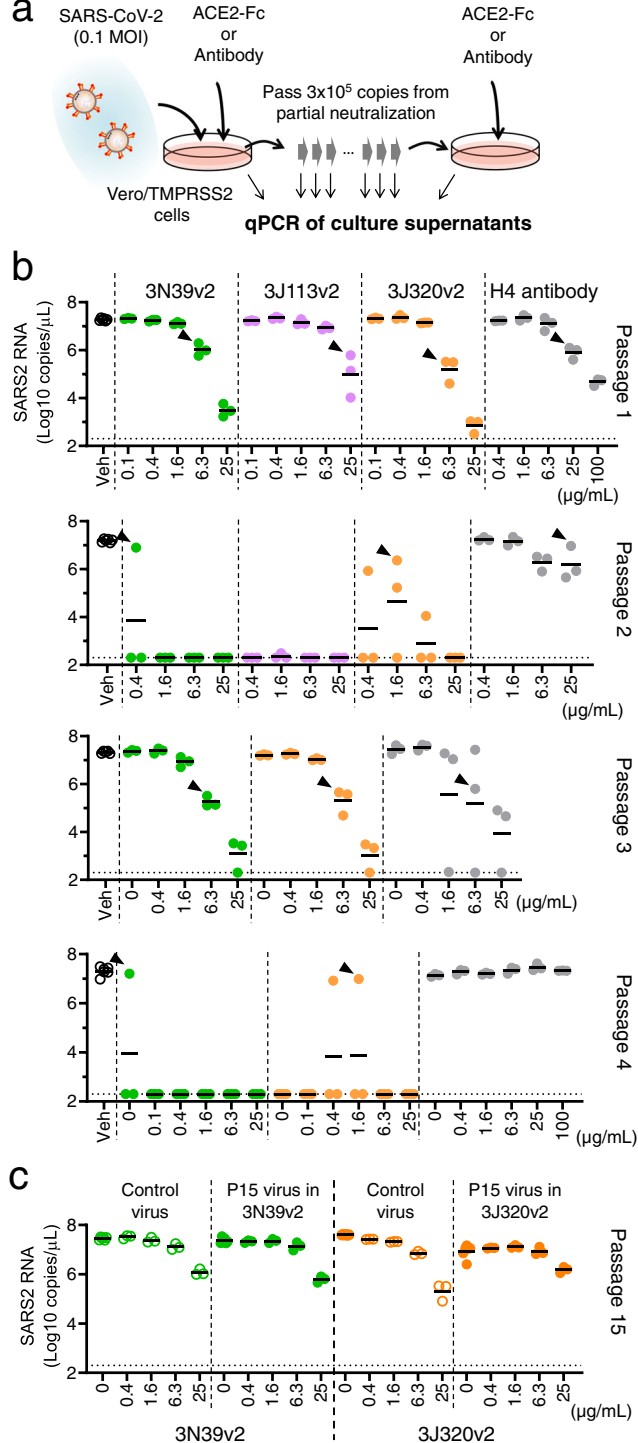

**Fig. 3 No emergence of escape mutation under the treatment of modified ACE2-Fc. a** Protocol of generating escape mutation in SARS-CoV-2. At first, 0.1 MOI of SARS-CoV-2 was cultured in Vero6E/TMPRSS2 cells with indicated concentration of ACE2-Fc or H4 antibody, then a total of 3 ×10⁵ copies of virus in partially neutralized well was passed in the presence of ACE2-Fc or H4 antibody dilution. Supernatants were collected from each well and analyzed virus RNA copy number by quantitative PCR. **b** The copy number of SARS-CoV-2 genome RNA in cultured medium was analyzed in each passage. The virus from indicated well (arrow head) was passed and escape mutant expansion was observed only in H4 antibody at passage 4. The test for 3J113v2 was discontinued due to no growth of virus at passage 2. **c** Viruses at passage 15 were neutralized similarly with the passaged control in ACE2-Fcs.

(Supplementary Fig. 9e). When this mutation was introduced in the context of either WT or its 3N39v2 version, their RBD-competing activity remained unchanged from those of the original proteins (Supplementary Fig. 9f). Furthermore, the same mutation did not result in the upregulation of RBD-binding affinity of the monomeric ACE2-His proteins (Supplementary Fig. 9g), indicating that the closed conformation per se is not responsible for the enhanced affinity. Interestingly, introduction of this disulfide bond did cause large increase in the stability of the ACE2 protease domain, indicated by the $T_m$ shift of +12.5, +7.0, and +11.5 degrees for WT, 3N39, and 3N39v2 ACE2, respectively (Fig. 2e), suggesting the potential benefit of this mutation in formulating the anti-COVID-19 therapeutics. Besides, this observation gives us an important lesson that the inhibitor-blocked ACE2 in the closed conformation can still engage SARS-CoV-2 RBD, suggesting that caution must be taken when considering the possibility of using ACE2 inhibitors as the therapeutics against coronavirus infection[29].

**No emergence of mutational escape.** There is a general concern in antiviral therapies that pathogenic virus could acquire drug resistance due to the frequent mutation. Actually, it was reported that SARS-CoV-2 escape mutation arose rapidly during the culture with a neutralizing antibody[11, 30]. We evaluated the occurrence of escape mutation using authentic SARS-CoV-2 virus. At the first passage, 0.1 MOI virus was added to the culture in the presence of serially diluted mutant ACE2-Fc or recombinant monoclonal antibody (clone H4) isolated from a convalescent patient[31], and a total of 3 ×10⁵ copies of amplified virus from partially neutralized well was transferred to the next passage (Fig. 3a). Consistent with previous report[11, 30], the virus pool after the treatment with single antibody became insensitive to the highest concentration of the same antibody after 4 passages (Fig. 3b) and sequencing of escape mutant revealed that F490V ablated the binding of H4 antibody. In contrast, the neutralizing capacity of the mutant ACE2-Fcs remained very high even against the virus pool after 15 passages, indicating the lack of emergence of escape mutations (Fig. 3c). We also confirmed that mutant ACE2-Fcs effectively neutralized H4-resistant F490V mutated virus, as well as escape mutants from COVID-19 convalescent plasma, Δ69-70/D796H that emerged in the immunosuppressed patient chronically treated with convalescent plasma[13] and insertion of N-linked glycan sequence (248aKTRNKSTSRRE248k) from the long-term cell culture experiment[12] (Supplementary Fig 10a-c). Recently, highly transmissible variants have emerged in the United Kingdom (B.1.1.7), South Africa (B.1.351), Brazil (P.1) and now they are global concerns. Among them, B.1.351 and P.1 carrying E484K mutation exhibits the potential escaping from natural and vaccine induced sera and monoclonal antibodies derived from them[32, 33]. ACE2-Fcs exhibited effective neutralization even against these variants in both pseudotyped (Fig. 4a) and authentic viruses (Fig. 4b).

**Therapeutic potential of engineered ACE2 decoy receptor.** Among three v2 mutant ACE2-Fcs showing similar neutralizing capacity and structural stability, we selected 3N39v2-Fc based on its high expression efficiency (Supplementary Fig. 11) to evaluate therapeutic potential in hamster model of COVID-19 characterized by rapid weight loss and severe lung pathology[34]. Since pharmacokinetic analysis of ACE2-Fc in mice indicated reasonable distribution in blood and lung tissues after intraperitoneal administration (Supplementary Fig. 12), 3N39v2-Fc or control-Fc were intraperitoneally administered to hamsters 2 hr after

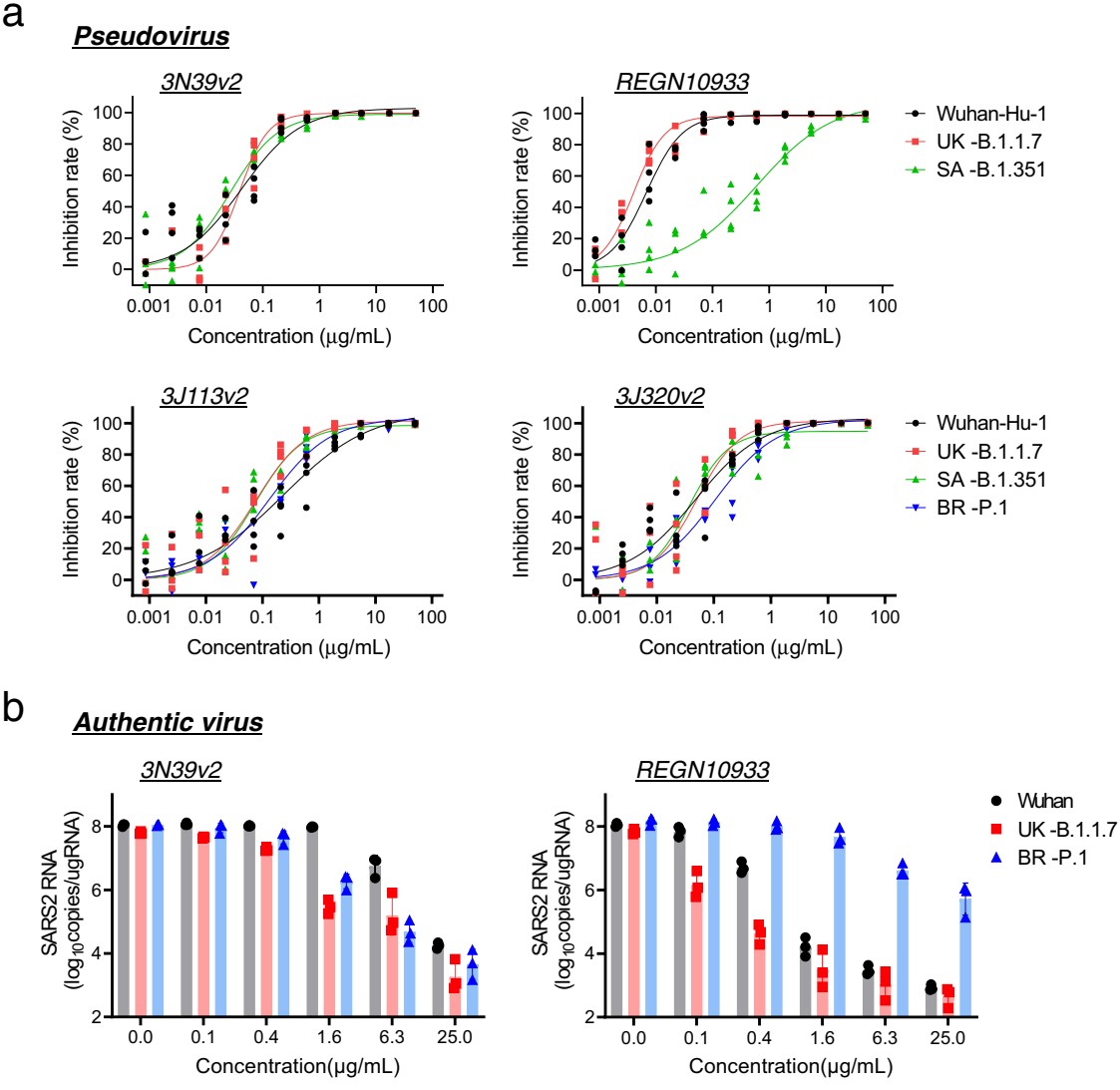

**Fig. 4 Preserved neutralizing effect in 3N39v2 against globally spreading SARS-CoV-2 variants. a** ACE2-Fcs and monoclonal antibody, REGN10933[46] neutralized pseudotyped variants recently emerged in the United Kingdom (B.1.1.7), South Africa (B.1.351), and Brazil (P.1) in 293T/ACE2 cells. $n = 4$ technical replicates. **b** Neutralization potency of 3N39v2-Fc and monoclonal antibody, REGN10933 against UK-B.1.1.7 and BR-P.1 was analyzed in Vero6E/TMPRSS2 cells. Data are mean ± SEM of $n = 3$ technical replicates.

intranasal $1.0 \times 10^6$ plaque forming units (PFU) virus challenge, as a model of therapeutic application (Fig. 5a). Hamsters that received control-Fc lost 4.3% of body weight, whereas those treated with 3N39v2-Fc gained 7.3% similarly to the unchallenged control 5 days post-infection (dpi) (Fig. 5b). In micro-CT before necropsy at 5 dpi, the control group showed multi-lobular ground glass opacity mainly in the cranial portion, conversely, lung abnormalities were limited in the treated one (Fig. 5c, Supplementary Movies 1–3). The content of SARS-CoV-2 in lungs was evaluated as functional virus particle and genome RNA copy number, and it was revealed that both were significantly decreased in 3N39v2-Fc-treated group (Fig. 5d). We also performed histopathological analyses of infected hamsters. The control hamsters showed severe interstitial pneumonia characterized by widespread infiltration of inflammatory cells, alveolar septal thickening, and alveolar hemorrhage, whereas 3N39v2-Fc treatment obviously reduced lung pathologies and viral antigen (Fig. 5e–f). Consistent with these results, the expression of proinflammatory or chemotactic cytokines in 3N39v2-Fc-treated lungs at 5 dpi showed decreased level of interleukin (IL)-6,

interferon (IFN)-γ, IFN-λ, CXCL10, CCL3, CCL5, as well as IL-10 and TGF-β that are also associated with COVID-19 severity[35, 36] (Fig. 5g, Supplementary Fig. 13). As the more practical setting of clinical use, we examined the therapeutic effect of the administration at 2 dpi. Analysis at 5 dpi showed modest but significant decreases in viral RNA copy and cytokine expression (Supplementary Fig. 14), which suggests a therapeutic potential of engineered ACE2 against human COVID-19.

In addition to working as a decoy receptor, ACE2-Fc might have a protective effect on acute lung injury through proteolysis of angiotensin II[15, 37]. However, it could have off-target vasodilation effect in the high-dose administration and actually our ACE2 mutants exhibited similar catalytic activity with WT in the assay of fluorogenic substrate cleavage (Supplementary Fig. 15). The closed-type disulfide mutation in the binding pocket of ACE2 exhibited no enzymatic activity (Supplementary Fig. 9e) with better structural stability, as well as intact binding affinity and neutralizing activity in both WT and 3N39v2 background (Supplementary Fig. 9f–g), which could lead to more effective and safer treatment.

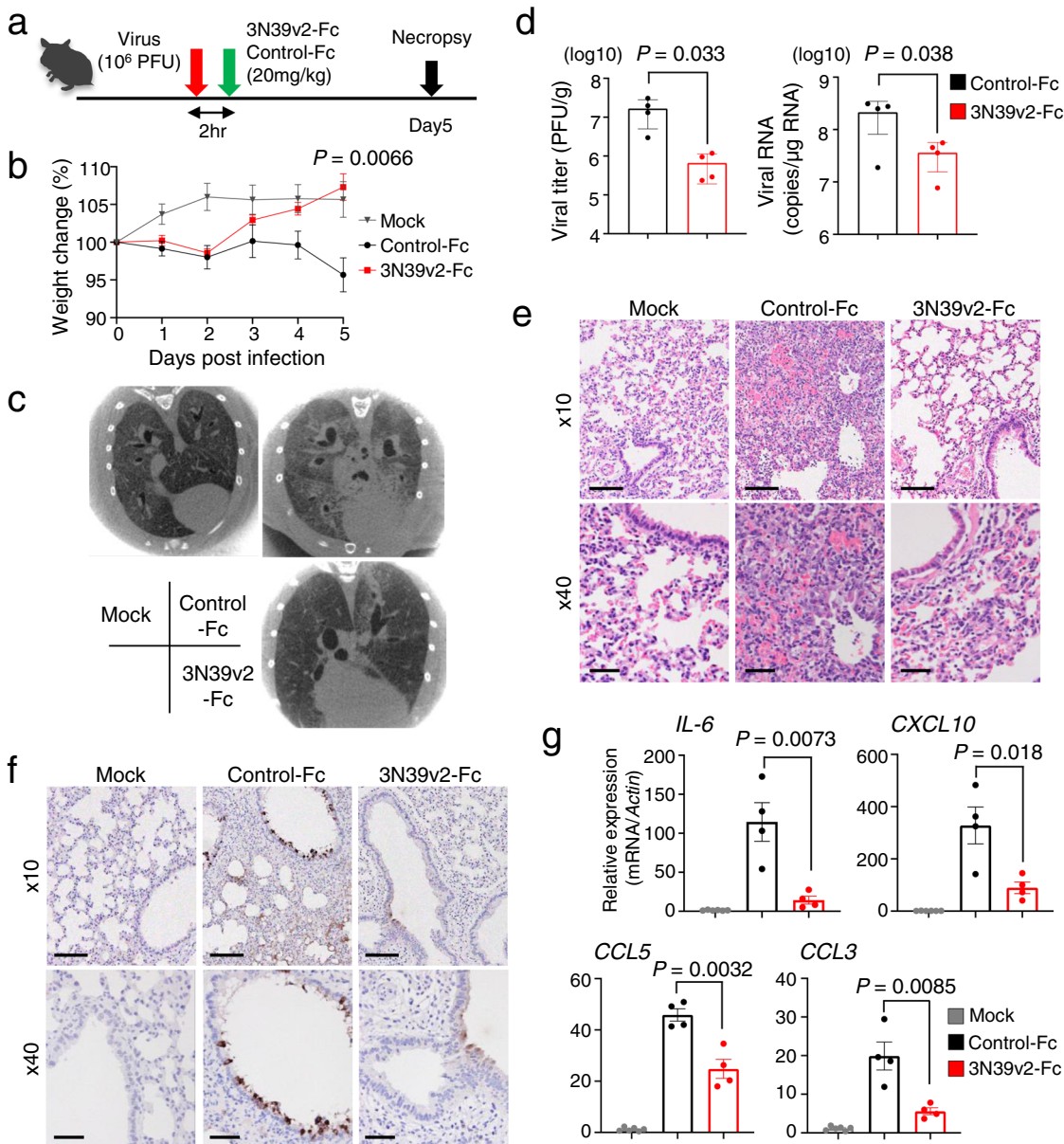

**Fig. 5 Therapeutic efficiency of 3N39v2-Fc in a COVID-19 Hamster Model. a** Schematic overview of the animal experiment. **b** Percent body weight change was calculated from day 0 for all hamsters. **c** Axial CT images of the thorax 5 days after infection. **d** Quantification of plaque-forming units (PFU) from lung homogenates and genomic SARS-CoV-2 RNA as copies per μg of cellular transcripts. **e** H&E staining and **f** SARS-CoV-2 antigen staining of hamster lung lobes. Scale bars, 100 μm (upper panel) and 40 μm (lower panel). **g** mRNA expression of inflammatory or chemotactic cytokines in hamster lung lobes. **b**, **d**, **g** Data are mean ± SEM of $n = 6$ for mock group, 4 for each treated group. $P$-values by two-sided unpaired $t$-test.

## Discussion

The pandemic of COVID-19 continues to expand and various mutants of SARS-CoV-2 had already emerged around the world. These mutants may not only increase infectivity and virulency but generate resistant strains to vaccines and therapeutic agents. To date, two neutralizing antibody drugs are clinically developed against COVID-19[5, 30]. However, there is concern about the early emergence of resistant strains when used as a single antibody and the risk is also reported even in the cocktail reagent[10]. High affinity modified ACE2 fused with Fc is the promising strategy to neutralize SARS-CoV-2 that can overcome the problem of drug-resistant strains. The risk of emerging mutational escape can be experimentally analyzed by co-incubation of SARS-CoV-2 with partially effective dose of drugs[11, 12, 30]. Replacement by escape mutants against single monoclonal antibody was observed after 4

passages in this study. It was also reported to happen after 14 passages even in the presence of COVID-19 convalescent plasma[12], whereas we found that engineered ACE2 induced no resistant virus after at least 15 passages.

We developed the screening system based on the cycle of random mutagenesis and sorting of high affinity population in 293T cells followed by validation of neutralizing activity in a soluble form. In this screening, an additional random mutation was induced in the bulk of sorted mutants, which worked better than mutagenesis in the top mutant. Engineering of decoy receptors with improved affinity was previously reported for cancer-related molecules and ACE2[18, 38, 39]. They used yeast display system to perform directed evolution. Large scale library (~$10^7$ mutants) was prepared and high affinity mutants were identified by repeating sorting from initial library. Fast growth

rate of yeast is suitable for library screening involving repeated sorting and propagation. On the other hand, we employed human cells for the display purpose. Since post-translational modification can modulate protein binding affinity, human cell-based screening is better to understand the impact of ACE2 variants on viral affinity and also to proceed biologics development. Repeating mutagenesis after cell sorting without propagation enabled us to conduct screening with relatively small library (~$10^5$ mutants) in human cells. During the validation, we noticed that high affinity pattern of full length ACE2 binding RBD-sfGFP did not always correlate with its RBD-competing activity in the flow cytometry. Thus, it is evident that experimental validation of each mutation at the level of sACE2 protein was important for efficient identification of high affinity mutants.

Several mutants of ACE2 with increased spike RBD affinity were also reported by Procko[17] and Wells[18]. These mutations, along with the current ones, are summarized in the Supplementary Table 2. Mutations are frequently found in the residues located at the interface with spike RBD including A25, K31, H34, E35, and L79, although the type of amino acid replacement were variable and no strong correlation patterns in the occurrence of particular mutations have emerged. Interestingly, mutations found in this study and Wells' paper are limited within PD1, whereas Procko's group found mutations both in PD1 and PD2. We envisioned that additional mutations in PD2 onto the initial PD1-limited mutants may enable discovery of better mutants, but no such mutants could be obtained (Supplementary Fig. 1b). We also tried to combine our 3N39v2 with individual affinity-enhancing PD2 mutations in Procko's deep mutagenesis panel[17], but none of them showed increased affinity but were rather detrimental (Supplementary Fig. 1e). The failure of the cumulative approach suggests that the contribution of each mutation to the increased affinity is highly context-dependent, pointing toward the need for obtaining structural information at atomic resolution for more mutants to instruct designs of even better therapeutic candidates. Another important point to be considered is the way of performing the selection. The SPR data suggest that promising mutants isolated in three studies (i.e., 3N39, 3J113, and 3J320 in ours, v2.4 in Procko[17], and #313 in Wells[18]) all show very slow dissociation rate from the spike RBD, suggesting that slowing down the dissociation further may not result in the higher FACS signal during the selection cycles, precluding the isolation of clones with even higher affinity. Therefore, selection methods orthogonal to the current ones may need to be combined to obtain further ACE2 mutants.

As the ACE2-Fc proteins were administered intraperitoneally in our mouse study, rather than intravenously, the pharmacokinetics could not be interpreted based on a simple one-phase decay. Assuming the two-phase PK model (i.e., the first phase representing the transfer of the agents from the peritoneal cavity to blood and the second phase representing the decay in the blood) and that the first phase would be completed after 24 h, only the plasma concentration values between day 1 to 7 were applied to one phase model, leading to the estimated plasma half-life of ~ 30 h for WT-Fc and ~ 25 h for 3N39v2-Fc. The half-life of rsACE2 (i.e., non-Fc fusion) was reported to be ~ 3 h in human subjects[23, 24], indicating that fusion with human IgG1 Fc prevented the fast drug disappearance. A recent paper showed that ACE2-Fc containing so-called LALA mutation exhibited much longer half-life of ~ 145 hours[40]. LALA mutation in the lower hinge abrogates FcγR binding while preserving the FcRn binding, which enables to recycle drugs and prolong its half-life. Engagement of FcγR through Fc is essential for the maturation of dendritic cells and to induce protective CD8+ T-cell responses[41], indicating that utilizing a non-LALA type of Fc with intact effector functions are required for the efficient therapeutic

protection in COVID-19 patients. On the other hand, for prophylactic purpose, FcRn-preserved LALA mutant Fc may be advantageous due to the longer effect[42].

One concern is the immunogenicity of ACE2 mutation in the step of drug development. Some biologics are notorious for their immunogenicity and inducing anti-drug antibodies. This may occur even with fully human monoclonal antibodies. In that case, there is a possibility that the patient will have the immune memory cross-reactive to the endogenous ACE2 that can in turn induce long-term adverse effects[43]. To avoid the undesired immune reaction, it will be crucial to reduce the dose and frequency of administration and to minimize the immunogenic mutation by employing various methods including in silico prediction and in vitro T cell activation assay[44].

The time frame for running one cycle of mutagenesis and sorting was just one week in our system, and we succeeded in developing optimized mutants in a couple of months independently of patients-derived cells or tissues. Thus, our system can rapidly generate therapeutic candidates against various viral diseases and may be well suited for fighting against future viral pandemics without a fear of mutational escape of viruses.

## Methods

**Cell culture**. Lenti-X 293T cells were purchased from Clontech and cultured at 37 °C with 5% $CO_2$ in Dulbecco's modified Eagle's medium (DMEM, WAKO) containing 10% fetal bovine serum (Gibco) and penicillin/streptomycin (100 U/ml, Invitrogen). VeroE6/TMPRSS2 cells were a gift from National Institutes of Biomedical Innovation, Health and Nutrition (Japan) and cultured at 37 °C with 5% $CO_2$ in DMEM (WAKO) containing 5% fetal bovine serum (Gibco) and penicillin/streptomycin (100 U/ml, Invitrogen). All the cell lines were routinely tested negative for mycoplasma contamination.

**Plasmids**. The mature polypeptide (a.a. 18-805) of human ACE2 (GenBank NM_021804.1) was cloned into the KpnI-XhoI sites of pcDNA4TO (Invitrogen) or the XbaI-SalI sites of pLenti puro with a N-terminal synthetic leader (MWWRLWWLLLLLLLLLWPMVWA)[45], HA-tag, and linker (GSGG). Some codon optimizations were introduced to generate BamHI and SacII and destroy original BamHI and SacII restriction sites. Soluble ACE2 (a.a. 18-615) fused to superfolder GFP (sfGFP) with a linker (GSGGSGSGGS) was placed between the BamHI-XhoI sites of the former plasmid. Equivalent sACE2 constructs were cloned with TEV protease cleavage site and 8 histidine tag or the Fc region of IgG1 into the KpnI-BstBI sites of pcDNA3.1-MycHis (Invitrogen) or the HindIII-BamHI sites of p3xFLAG-CMV-14 (Invitrogen). A codon-optimized RBD (a.a. 333-529) of SARS-CoV-2 Spike fused to sfGFP was obtained from addgene #141184[17] and cloned into the KpnI-XhoI sites of pcDNA4TO (Invitrogen) with a N-terminal synthetic leader, HA-tag, and linker (GSGG).

**Protein production and purification**. sACE2-His, sACE2-Fc, and RBD-Fc were expressed using the Expi293F cell expression system (Thermo Fisher Scientific) according to the manufacturer's protocol. His-tagged and Fc-fused proteins were purified from conditioned media using the Ni-NTA agarose (QIAGEN) and the rProtein A Sepharose Fast Flow (Cytiva), respectively. Fractions containing target proteins were pooled and dialyzed against phosphate buffered saline (PBS). The full-length spike trimer protein was also produced in Expi293F cells by stably transfecting expression vector coding for the entire ectodomain portion of SARS-CoV-2 spike protein (residues 1-1212 with stabilizing mutations R684G, R685S, R678G, K998P, and V999P) with a C-terminal fibritin trimerization motif and hexahistidine tag (a gift from Takao Hashiguchi, Kyushu University). The spike trimer was purified from the culture supernatants by using Ni-NTA agarose followed by dialysis against PBS. An anti-RBD monoclonal antibody (clone H4) isolated from a convalescent patient[31] and REGN 10933 antibody[46] were formulated in the form of human IgG1/kappa by using synthetic DNA coding for the variable regions of heavy and light chains taken from the publicly available amino acid sequences, and recombinantly produced in Expi293F cells as above.

**Library preparation**. Error-prone PCR was performed in ACE2 residues 18-102 and 272-409 independently using GeneMorph II Random Mutagenesis Kit (Agilent). A 0.1 ng of plasmid was used as a PCR template and generated mutations with an average of about one mutation per 100 bp in 35-cycle reaction. The plenti ACE2 vector was digested with BamHI or MfeI-SacII (NEB) for ACE2 residues 18–102 or 272–409, respectively with alkaline phosphatase (Fermentas) at 37 °C for 2 h and gel-purified on a Gel and PCR Clean-up kit (TAKARA). A 160 µl NEBuilder ligation reaction (NEB) was performed using 5 ng of the gel-purified inserts and 10 ng of the vector, then purified on a Gel and PCR Clean-up kit (TAKARA)

and eluted by a 20 μl of distilled water. From the ligation, 400 μg (~8 μl) of the purified reaction was transformed into 100 ul of electrocompetent cells (Lucigen) and expanded according to the manufacturer's protocol with 1500 V electroporation by ECM 399 (BTX). A 1000-fold dilution of the full transformation was plated to estimate the scale of mutant library.

**Virus production**. Ten-centimeter plates of 70% confluent Lenti-X 293T (Clontech) cells were transfected with 9 μg of the plasmid library, 6 μg of psPAX2 vector and 3 μg pMD2.G using Fugene HD (Promega) according to the manufacturer's instructions. Supernatant was collected after 48 h and then spun for 10 min at 4 °C at 3,000 rpm and then filtered with a 0.45 μm low protein-binding filter (SFCA), and frozen at –80 °C. The titers of the virus was determined by using 293T cells followed by puromycin selection.

**Library screening**. The mutant library was transduced into 293T cells via spinfection. To find optimal virus volumes for achieving a multiplicity of infection (MOI) of 0.1–0.3, each library virus was tested by spinfecting $5\times10^5$ cells with several different volumes of virus in the 6-well plate. Each well received a different titrated virus amount (usually between 50 and 500 μl) along with a no-transduction control and infected cell rate was determined by anti-HA Alexa 594. Then, four 6-well plates, $1.2\times10^7$ cells were centrifuged at 1000 g for 1.5 h at 37 °C.

Cells were sorted using a SH800 (SONY) 24 h after spinfection. Approximately $5\times10^7$ induced library cells were resuspended with complete medium and incubated for 30 min at 4 °C with a 1/40~1/160 dilution of medium containing RBD-sfGFP and a 1/4000 dilution of anti-HA Alexa 647 (clone TANA2, MBL). Cells were directly sorted on SH800 (SONY). The top 0.05% of cells were sorted and their genomic DNA was extracted by NucleoSpin Tissue (TAKARA). Mutated ACE2 fragment was cloned into pcDNA4TO ACE2 for individual validation and also introduced random mutations again with error-prone PCR for further screening.

**Flow Cytometry Analysis**. The pcDNA4TO HA-ACE2 plasmid was transfected into 293T cells (500 ng DNA per ml of culture at $5\times10^5$ / ml) using Fugene HA (Promega). Cells were analyzed by flow cytometry 24 h post-transfection. To analyze the binding of RBD-sfGFP to full length HA-ACE2, cells were washed, trypsinized, and resuspended with complete medium, then incubated for 30 min at 4 °C with a 1/40~1/160 dilution of medium containing RBD-sfGFP and a 1/1000 dilution of anti-HA Alexa 594 (clone TANA2, MBL). Cells were directly analyzed on Attune NxT Flow Cytometer (Invitrogen).

To analyze the competitive activity of sACE2-sfGFP against RBD-sfGFP, a serial dilution of medium containing sACE2-sfGFP and a 1/40 dilution medium containing RBD-sfGFP were mixed and incubated for 1 hr at 4 °C, then HA-ACE2 expressing 293T cells were resuspended and incubated with the mixture and a 1/1000 dilution of anti-HA Alexa 594 (clone TANA2, MBL) for 30 min at 4 °C. Cells were directly analyzed on Attune NxT Flow Cytometer (Invitrogen).

The competitive activity of ACE2-Fc against soluble spike trimer binding to cell-surface ACE2 was evaluated as follows. First, varying concentrations (0.3–30 nM) of purified spike trimer was preincubated with 60 μg/ml (~315 nM) ACE2-Fc proteins at room temperature for 2 hr, followed by addition to Expi293F cells transiently transfected with human full-length ACE2 2 days before the experiment. After incubation on ice for 1.5 h, the cells were washed 3 times with PBS and further incubated with a 1/1000 dilution of Alexa Fluor 488-labeled anti-His tag antibody (MBL, catalog # D291-A48) at 2 μg/ml to stain the bound spike protein. Cells were analyzed on an EC800 system (Sony) and the data were processed with FlowJo (FlowJo, LLC).

**Kinetic binding measurement using Biacore (SPR)**. The binding kinetics of sACE2 (wild-type or mutants) to RBD were analyzed by SPR using a Biacore T200 instrument (Cytiva) in a single-cycle kinetics mode. Anti-human IgG (Fc) antibody was immobilized onto a CM5 sensor chip (Cytiva) using the Human Antibody Capture Kit (Cytiva) according to the method provided by the manufacturer. The RBD-Fc was captured on the measurement cell via the antibody at a density of ~450 RU, while human IgG1-Fc was captured on the reference cell at a density of ~200 RU. The binding was evaluated by injecting various concentrations of sACE2-His solutions in series using PBS containing 0.05% Tween 20 as a running buffer. The runs were conducted at 25 °C employing the following parameters; flow rate of 30 μl/min, contact time of 120 s, and dissociation time of 480 s. After each run, the surface was regenerated by injecting the Regeneration solution contained in the kit for 30 s. The binding curves of the measurement cell were subtracted with those of the reference cell, and used to derive kinetic binding values. The results were evaluated by using Biacore T200 evaluation software version 4.1.

**Thermal shift assay**. The thermal stability of the mutant ACE2 ectodomain proteins were evaluated by differential scanning fluorimetry as follows. Purified ACE2-His proteins were diluted to 200 μg/ml in PBS and placed in 0.2-mL white PCR tubes (Bio-Rad, TLS0851) at 20 μl/tube. After adding 1 μl/tube of SYPRO™ Orange protein gel stain solution (Invitrogen, S6651) diluted with water at 1:150, the tubes were placed in a Bio-Rad CFX96 thermal cycler Real-Time Detection System. Thermal denaturation curves from 25 °C to 95 °C (ramp rate of

1.27 °C/min at 0.5 °C intervals with an equilibration of 5 sec at each temperature before measurement) were acquired by measuring fluorescence intensities using the FRET channel with excitation from 450 to 490 nm and detection from 560 to 580 nm. All data were exported and plotted in Microsoft Excel and the first derivative approach was used to calculate $T_m$.

**Mass Photometry**. Binding stoichiometry between ACE2-His and spike trimer was determined by mass photometry[47]. To this end, either WT or 3N39v2 mutant ACE2-His (~75 kDa polypeptide with 5 N-glycans) was incubated with purified SARS-CoV-2 spike trimer (~450 kDa polypeptide with 63 N-glycans) at 1: 1.2 molar ratio in PBS and placed on a microscope coverslip. MP data were acquired and analyzed using a One$^{MP}$ mass photometer (Refeyn Ltd, Oxford, UK).

**Crystallization**. For crystallization, hingeless Fc was appended to RBD (RBD-noHg-Fc), and the protein was expressed in Expi293F cells in the presence of 5 μM α-mannosidase inhibitor, kifunensine (Cayman Chemical Co.). After purification using the rProtein A Sepharose, RBD-noHg-Fc was treated with His-tagged IdeS protease to cleave between the RBD and the Fc, followed by subjecting to the Ni-NTA agarose and the rProtein A Sepharose sequentially to remove His-IdeS and Fc, respectively, by recovering the unbound fractions. The sample was further purified by size-exclusion chromatography (SEC) on a Superdex 200 Increase 10/300 GL column (Cytiva) equilibrated with 20 mM Tris, 150 mM NaCl, pH 7.5. 3N39 mutant sACE2-His was also expressed in Expi293F cells in the presence of kifunensine. After Ni-NTA purification, the sample was purified by anion exchange chromatography on a MonoQ 5/50 GL column equilibrated with 20 mM Tris, pH 8.0. To obtain sACE2-RBD complex, the purified sACE2 and RBD were mixed at a molar ratio of 1: 1.3 and subjected to SEC. Fractions containing the complex sample were concentrated to 9.2 mg/ml prior to crystallization. The diffraction quality crystal was grown under the condition of 1.6 M ammonium sulfate, 0.25 M lithium sulfate, and 0.05 M CAPS pH 10.5. The crystal was cryoprotected with the same buffer containing 25% ethylene glycol and used for data collection.

**Data collection, phasing, and structure refinement**. X-ray diffraction experiment was performed at beamline BL44XU of SPring-8 (Hyogo, Japan). Four datasets were collected at 100 K from a single crystal and were combined, processed, and scaled using the X-ray Detector Software[48]. Initial phase was determined by molecular replacement method with PHASER v.2.8.1[49] from the CCP4 package v.7.0[50] using the crystal structure of RBD-sACE2(WT) (PDB: 6m0j) as a search model. The structural model was modified with COOT v.0.8.9[51] and refined with PHENIX v.1.14[52]. Ramachandran plot analysis of the final structure with MOLPROBITY v.4.5[53] showed that 96.5 and 3.3% of the residues are in favored and allowed regions, respectively. Data collection, processing, and refinement statistics are summarized in Table 1. Data of wild-type ACE2/SARS-CoV-2 S-RBD complex, wild-type ACE2/SARS-CoV-1 S-RBD complex and inhibitor-bound closed conformation of ACE2 are derived from PBD codes: 6M0J (DOI: 10.2210/pdb6M0J/pdb)[26], 2AJF (DOI: 10.2210/pdb2AJF/pdb)[54] and 1R4L (DOI: 10.2210/pdb1R4L/pdb), respectively. All structural figures were prepared with the PyMOL software v.2.3.4. (https://pymol.org/2/).

**ACE2 catalytic activity assay**. Activity was measured using the Fluorometric ACE2 Activity Assay Kit (AnaSpec) with protein diluted in assay buffer to 100 ng/ml final concentration. Specific activity is reported as pmol MCA produced per 100 ng of enzyme. Fluorescence was read at 5 min interval on SpectraMax M2 (Molecular Devices).

**In-house sandwich ELISA for the quantification of ACE2**. Rabbit antisera were obtained by immunizing purified WT sACE2-His, and high-titer antisera were subjected to an immunoaffinity isolation using 3N39 mutant sACE2-Fc protein immobilized on Sepharose beads. The ACE2-specific IgG was directly labeled with horseradish peroxidase (HRP) by using Peroxidase Labeling Kit-NH2 (Dojindo Molecular Technologies, Inc., #LK11) according the method provided by the manufacturer. ACE2-Fc proteins in 100-fold diluted blood samples were captured onto microtiter plate (NUNC-Immuno plate #442404, Thermo Fisher) through goat anti-human IgG1 Fc F(ab')$_2$ (Sigma-Aldrich, I3391) by incubating at room temperature for 2 h, followed by detection with HRP-labeled anti-ACE2 IgG diluted at 0.4 μg/ml. Absorbance data after the addition of the peroxidase substrate ABTS (2,2'-Azinobis [3-ethylbenzothiazoline-6-sulfonic acid]-diammonium salt) were analyzed by Prism software to derive concentration of Fc fusion proteins by using standard curves obtained with corresponding ACE2 mutants at concentration range of 1-729 ng/ml in the same experiment.

**Single-dose pharmacokinetics of ACE2-Fc**. All animal experiments were performed according to procedures approved by Kyoto Prefectural University of Medicine Institutional Animal Care and Use Committees. Eight weeks old male C57BL/6JJcl mice were purchased from CLEA Japan and had free access to food and water and were caged at 23 °C with 12-h light cycles and normal humidity (40–70%). A single 20 mg/kg dose of ACE2-Fc was intraperitoneally injected and whole blood was harvested from tail veil at indicated time points sequentially.

Another cohort of mice was euthanized and whole blood and lungs were isolated 2 h after injection. Heparinized whole blood was centrifuged, and plasma was collected and frozen. Lungs were perfused with 10 mL of cold PBS to remove blood before harvest, and then lung tissue lysates were prepared through mechanical disruption using Precellys tissue homogenizer (Bertin), followed by lysis with Cell Extraction Buffer PTR (Abcam). Lysates were cleared by centrifugation and were frozen for analysis. ACE2-Fc concentration was analyzed using human ACE2 ELISA Kit (Abcam; ab235649) for lung lysates or in-house sandwich ELISA described above.

**Pseudotyped virus neutralization assay.** Pseudotyped reporter virus assays were conducted as previously described[55]. In brief, a plasmid coding SARS-CoV-1 Spike was kindly gifted from Takao Hashiguchi (Kyushu University) and SARS-CoV-2 Spike was obtained from addgene #145032[16], and deletion mutant CΔ19 (with 19 amino acids deleted from the C terminus) was cloned into pcDNA4TO (Invitrogen) to enhance virus titer[56]. Spike-pseudovirus with a luciferase reporter gene was prepared by transfecting plasmids (CΔ19, psPAX2, and pLenti firefly) into Lenti-X 293T cells with Lipofectamine 3000 (Invitrogen). After 48 h, supernatants were harvested, filtered with a 0.45 μm low protein-binding filter (SFCA), and frozen at −80 °C. ACE2-expressing 293T cells were seeded at 10,000 cells per well in 96-well plate. Pseudovirus and three-fold dilution series of sACE2-Fc protein were incubated for 1 h, then this mixture was administered to ACE2-expressing 293T cells. After 1 h pre-incubation, medium was changed and cellular expression of luciferase reporter indicating viral infection was determined using ONE-Glo™ Luciferase Assay System (Promega) in 48 h after infection. Luminescence was read on Infinite F200 pro system (Tecan).

**Viruses.** SARS-CoV-2 strains, 2019-nCoV/Japan/TY/WK-521/2020 (Wuhan lineage), hCoV-19/Japan/QHN002/2021 (UK-B.1.1.7 lineage), hCoV-19/Japan/TY7-503/2021 (BR-P.1 lineage) were isolated at National Institute of Infectious Diseases (NIID). SARS-CoV-2 were propagated in VeroE6/TMPRSS2 cells cultured at 37 °C with 5% CO₂ in DMEM (WAKO) containing 10% fetal bovine serum (Gibco) and penicillin/streptomycin (100 U/ml, Invitrogen). The virus stock was generated by infecting VeroE6/TMPRSS2 cells at an MOI of 0.1 in DMEM containing 10% FBS; viral supernatant was harvested at 2 days post infection and the viral titer was determined by plaque assay.

**SARS-CoV-2 neutralization assay.** Vero-TMPRSS2 were seeded at 80,000 cells in 24 well plates and incubated for overnight. Then, SARS-CoV-2 was infected at MOI of 0.1 together with sACE2-Fc protein. After 2 h, cells were washed by fresh medium and incubated with fresh medium for 22 h. Culture supernatants were collected and performed qRT-PCR assay.

**Syrian hamster model of SARS-CoV-2 infection.** All animal experiments with SARS-CoV-2 were performed in biosafety level 3 (ABSL3) facilities at the Research Institute for Microbial Diseases, Osaka University. Animal Experimentation, and the study protocol was approved by the Institutional Committee of Laboratory Animal Experimentation of the Research Institute for Microbial Diseases, Osaka University (R02-08-0). All efforts were made during the study to minimize animal suffering and to reduce the number of animals used in the experiments. Four weeks-old male Syrian hamsters were purchased from SLC Japan. Syrian hamsters were anaesthetized by intraperitoneal administration of 0.75 mg/kg medetomidine (Meiji Seika), 2 mg/kg midazolam (Sandoz) and 2.5 mg/kg butorphanol tartrate (Meiji Seika) and challenged with 1.0 ×10⁶ PFU (in 60 μL) via intranasal routes. After 2 h post infection, Control-Fc (20 mg/kg) or ACE2-Fc (3N39v2, 20 mg/kg) were dosed through intraperitoneal injection. Body weight was monitored daily for 5 days.

On 5 days post infection, all animals were euthanized and lungs were collected for histopathological examinations, virus titration and qRT-PCR. The structures of lungs were also observed by using micro computed tomography (μCT) (ScanXmate-RB080SS110, Comscantecno Co.,Ltd.) with the following parameters: source voltage 80 kV; current, 0.1 mA; and voxel size, 0.1 mm. The images of μ CT was reconstructed and visualized by ImageJ software.

**SARS-CoV-2 virus plaque assays.** VeroE6/TMPRSS2 cells were seeded on 24 well plates (80,000 cells/well) and incubated for overnight. The lung homogenates serially diluted by medium were inoculated and incubated for 2 h. Culture medium was removed, fresh medium containing 1% methylcellulose (1.5 mL) was added, and the culture was further incubated for 3 days. The cells were fixed with 4% Paraformaldehyde Phosphate Buffer Solution (Nacalai Tesque) and plaques were visualized by using a Crystal violet or Giemsa's azur-eosin-methylene blue solution (Merck Millipore: 109204).

**Quantitative RT-PCR.** Total RNA of lung homogenates was isolated using ISOGENE II (NIPPON GENE). Real-time RT-PCR was performed with the Power SYBR Green RNA-to-CT 1-Step Kit (Applied Biosystems) using a AriaMx Real-Time PCR system (Agilent). The relative quantitation of target mRNA levels was performed by using the 2-ΔΔCT method. The values were normalized by those of the housekeeping gene, β-actin. The primers were listed in Supplementary Table 3.

**Quantitative RT-PCR of Viral RNA in the supernatant.** The amount of RNA copies in the culture medium was determined using a qRT-PCR assay as previously described with slight modifications[57]. In brief, 5 μl of culture supernatants were mixed with 5 μl of 2× RNA lysis buffer (2% Triton X-100, 50 mM KCl, 100 mM Tris-HCl [pH 7.4], 40% glycerol, 0.4 U/μl of Superase•IN [Life Technologies]) and incubate at room temperature for 10 min, followed by addition of 90 μl of RNase free water. 2.5 μl of volume of the diluted samples was added to 17.5 μl of a reaction mixture. Real-time RT-PCR was performed with the Power SYBR Green RNA-to-CT 1-Step Kit (Applied Biosystems) using a AriaMx Real-Time PCR system (Agilent).

**Escape mutation study.** VeroE6/TMPRSS2 cells were seeded on 96 well plates and infected with SARS-CoV-2 at MOI of 0.1 together with recombinant proteins. After 2 days post-infection, culture supernatants were collected, quantified copy numbers of viral RNA and a total of 3 ×10⁵ copies of virus were further infected in naïve VeroE6/TMPRSS2 cells with recombinant proteins.

**Hematoxylin and eosin staining and Immunohistochemistry.** Lung tissues were fixed with 10% neutral buffered formalin and embedded in paraffin. 2 μm tissue sections were prepared and stained with Hematoxylin and eosin (H&E). For immunohistochemical staining, 2 μm thickness sections were immersed in citrate buffer (pH 6.0) and heated for 20 min with a pressure cooker. Endogenous peroxidase was inactivated by immersion in 3% H₂O₂ in PBS. After treatment with 5% BSA in PBS for 30 min at room temperature, the sections were incubated with mouse anti-Spike protein antibody (1:300, clone 1A9, Genetex, Irvine, CA, USA). For secondary antibody, EnVision+ system-HRP-labeled polymer anti-mouse or anti-rabbit (Dako, Carpinteria, CA, USA) were used. Positive signals were then visualized by peroxidase–diaminobenzidine reaction and sections were counterstained with hematoxylin.

**Reporting summary.** Further information on research design is available in the Nature Research Reporting Summary linked to this article.

## Data availability
Crystallographic coordinates and structure factors for 3N39-RBD complex have been deposited in the Protein Data Bank under accession code 7DMU (DOI: 10.2210/pdb7DMU/pdb). Structural data of wild-type ACE2/SARS-CoV-2 S-RBD complex, wild-type ACE2/SARS-CoV-1 S-RBD complex and inhibitor-bound closed conformation of ACE2 are derived from PBD codes: 6M0J (DOI: 10.2210/pdb6M0J/pdb), 2AJF (DOI: 10.2210/pdb2AJF/pdb) and 1R4L (DOI: 10.2210/pdb1R4L/pdb), respectively. The source data underlying Figs. 1d, 3b,c, 4a,b, 5b,d,g, and Supplementary Figs 1c,e, 2b-g, 7, 9e,f, 10a-c, 12a,b, 13, 14b,c and 15 are provided as a Source Data file. Source data are provided with this paper.

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

## Acknowledgements

We would like to thank Sho Hashimoto, Toshiyuki Nishiji, Tomohiro Hino, and Keiko Tamura-Kawakami for the construction of expression vectors; Yukiko Takemura for production of ACE2 proteins; Tomoya Kitani and Daisuke Ueno for the validation experiment; Shunta Taminishi for NGS sample preparation; Takeshi Yaoi for helpful discussion and support; Takao Hashiguchi for kind gift of plasmids coding for SARS-CoV-1 and 2 Spike protein. This work was supported by Japan Agency for Medical Research and Development (AMED), Platform Project for Supporting Drug Discovery and Life Science Research (Basis for Supporting Innovative Drug Discovery and Life Science Research) under JP20am0101075 (J.T.), Research Program on Emerging and Re-emerging Infectious Diseases under JP20fk0108263 (T.O.) and JP20fk0108296h0001 (A.H., J.T. and T.S.) and grant from SENSHIN Medical Research Foundation (A.H.).

## Author contributions

A.H. designed the research; Y.H., N.I. and A.H. performed the directed evolution screening; Y.H. performed RBD neutralization assay and analyzed pharmacokinetics; N.I. performed pseudovirus neutralization assay and ACE2 catalytic activity assay; Y.K., D.M. and S.N. performed and analyzed next-generation sequencing; T.A., E.M. and J.T. purified and prepared the proteins; T.A. performed Biacore assays, Mass photometry, and X-ray structure analysis; E.M. conducted neutralization assay for S trimer; J.T. performed thermal shift assay; E.O. performed SARS-CoV-2 neutralization assay; T.S., Y.I., F.S. and T.O. conducted SARS-CoV-2 experiments in the cell culture and hamsters; Y.S. performed histological analysis; O.M., Y.M., S.M., T.O., J.T. and A.H. supervised the research; A.H., J.T. and T.O. wrote the manuscript; all authors discussed the results and commented on the manuscript.

## Competing interests

The authors declare no competing interests.
