## [Peer Review File · Nature Communications]

REVIEWER COMMENTS

Reviewer #1 (Remarks to the Author):

In this manuscript the authors used human cell-based directed evolution to select for an ACE2 protein with tighter binding to the RBD of the spike protein. After three cycles of random mutagenesis they obtained a 100-fold tighter binding ACE2 variant. Using SARS-CoV-2 they showed that no escape mutant was developed after 15 passages. This point is important, particularly now, that escape mutants take hold. They solved the x-ray structure of the complex, and showed that the high affinity ACE2 is protective to hamster, if given at a dose of 20 mg/Kg, 2 hr after their infection with CoV-2. This is a well performed and nice study on a potential drug against COVID-19.

The major problem here is with novelty. A number of studies were published, using similar strategies. For example, Wells et al published in PNAS (doi.org/10.1073/pnas.2016093117) a similar study, optimizing ACE2 for spike binding. About the same time, Procko published in science a study of ACE2 optimization for the same purpose (DOI: [10.1126/science.abc0870](https://doi.org/10.1126/science.abc0870)). Dang published a study on making an optimized ACE2 trimer, with super high neutralization efficacy towards CoV2 (Cell Research volume 31, pages98–100(2021). In the above cited studies, the advantage of using a decoy receptor, as it avoids virus escape mutants is also discussed. In addition to these, other studies using similar strategies (on the basis of ACE2 optimization) have been deposited in bioRxiv.

Specific comments:

1. The author referred to the potential of the spike to bind 3 ACE2 proteins. It would be appropriate here to cite the work Cell Research volume 31, pages98–100(2021), where this has been done (including a structure).
2. An interesting point to consider is the different mutations discovered in this study, versus other studies where ACE2 was evolved to bind tighter to the RBD. The authors should discuss this issue, as it may allow for even higher binding affinity to be generated. Computational mutant scanning has indeed showed that some of the differences in the residues mutated in the different studies can be attributed to the non-optimal receptor amino acids at some positions within the ACE2-spike interface.
3. In the animal experiment a very high amount of ACE2-FC was given (20 mg/Kg, IP). I would be worried that this high amount of mutant ACE2 may result in generation of antibodies against ACE2, which could be harmful for the future for the patient. Was this evaluated? Despite the high dose of ACE2, the therapeutic effect (reduction of viral titer and cytokine secretion) was significant, but smaller than expected. Why not use inhalation of a much lower dose? What would happen if the drug would be given 2 days after virus inoculation, to better mimic treatment in human. Will there still be a therapeutic effect?

Reviewer #2 (Remarks to the Author):

The manuscript entitled: "Engineered ACE2 receptor therapy overcomes mutational escape of SARS-CoV-2" by Higuchi et al. describes how sACE2 can be used therapeutic lead to treat COVID-19 patients. The idea for using sACE2 as COVID-19 treatment is not new however, the authors developed improved therapeutic leads with higher affinity and extended half-life.

The research group started with mutating and screening for sACE2 variants that exhibit nM affinities to the SARS-CoV-2 receptor binding domain. The focused-on mutation at the spike protease domains (PD) as it is known to harbor the interface to viral spike protein. The screening was carried out with a reporter utilizing HEK293 cells. High-affinity sACE2 was further improved by identification of essential mutations and the discovered variants were designated as version 2 of the leads.

The high-affinity sACE versions exhibited also very slow off-rate compared to the WT ACE2.

Mass photometry analysis indicated that the mutants can saturate all three RBDs on the SARS-CoV-2 spike trimer at low concentration, a property highly desirable for the virus-neutralizing agent.

Next, to reduce the fast clearance of the lead, they fused the mutated version of sACE to IgG1 Fc and evaluated their neutralization capacity against the SARS-CoV-2 in TMPRSS2-expressing VeroE6 cells and found that each mutant ACE2-Fc demonstrated a significant neutralizing effect even in 100-fold lower concentration than WT.

By structure analysis, the authors show that mutations collectively lead to the 100-fold increase in the overall affinity characterized by their binding affinities by surface plasmon resonance (SPR). The authors also learned from the structural data that the inhibitor-blocked ACE2 in the closed conformation can still engage SARS-CoV-2 RBD, suggesting that caution must be taken when considering the possibility of using ACE2 inhibitors as the therapeutics against coronavirus infection.

They further confirmed that the engineering sACE-Fc fusion does not generate escape mutants after 15 passages and also confirmed that mutant ACE2-Fcs effectively neutralized H4-resistant F490V mutated virus, as well as escape mutants from COVID-19 convalescent plasma, Δ 69-70/D796H6, and insertion of N-linked glycan sequence.

Finally, they selected the 3N39v2-Fc mutant based on its high expression efficiency to evaluate the therapeutic potential in a hamster model of COVID-19 characterized by rapid weight loss and severe lung pathology. They show that hamsters that received control-Fc lost 4.3% of body weight, whereas those treated with 3N39v2-Fc gained 7.3% similarly to the unchallenged control 5 days post-infection. They also analyzed the content of SARS-CoV-2 in lungs as functional virus particle and genome RNA copy number, and it was revealed that both were significantly decreased in 3N39v2-Fc-treated group.

Overall, the methods and experimental setups are solid and the data is convincing. The authors utilized several orthogonal approaches to well-characterize the discovered biological lead. However, there are two main concerns regarding the potential of such biologics to treat COVID-19 patients in the "real world".

First, what will be the effective dosage to be given to COVID-19 patients? given there is inter-patient variability of viral loads, what will be the effective dose to be used? and what will be the administration regimen, i.e. how many times will a patient need to be treated with the engineered sACE2-Fc so it will show high efficacy?

Derived from the first concern, will the treatment strategy (i.e. dosage and a number of administrations) elicit an immune response in the recipient. Some biologics are notorious for their immunogenicity and inducing anti-drug antibodies. This may occur even with monoclonal antibodies that are fully human. In the case of the engineered sACE2 and with dependence on the dosage strategy, there is a possibility that the drug will mount an anti-drug immune response by breaching tolerance (as it was mutated and cannot be considered WT). In such an event, there is a danger that the patient will be generated immune memory to the self-ACE2 that can in-turn induce long-term adverse effects. It will be good if the authors address both of these concerns.

Reviewer #3 (Remarks to the Author):

The Authors have provide evidence that engineered ACE2 have therapeutic potential in hamster animal model beyond the escape mutant SARS-CoV-2.

Major comments:

* The Authors need to include virus of concerns (VOC) such as 501y.v1, 5017.v2 and p1 in their animal model.

* The Author have used VeroE6 cells to push for escape mutant SARS-CoV2. The Authors need also include other cell lines and human primary cell lines to study the escape mutant.

* the Authors should include more Cytokines in figure 4.

* It is of interest to analyse the therapeutic effect of ACE2 at different time post infection. It is not practical to treat patient just after infection. the antiviral need to be used after symptoms appear.

Reviewer #4 (Remarks to the Author):

Review for

"Engineered ACE2 receptor therapy overcomes mutational escape of SARS-CoV-2"
Higuchi, Suzuki, Arimori, Ikemura et al.

In this nice study, the authors establish a mammalian display platform for directed evolution of ACE2 to build ACE2-Fc receptor decoys as therapeutic candidates to treat SARS-CoV-2 infections. They demonstrate neutralization efficacy of several of their candidates in pseudoviral and authentic SARS-CoV-2 viral neutralization cell culture assays. The mechanistic basis for increased affinity for spike binding in the evolved ACE2 candidate with the lowest IC50 is revealed by an X-ray crystal structure of the complex. The authors perform an experiment to compare the mutational escape of SARS-CoV-2 when treated with a neutralizing antibody to their receptor trap, which gives the impressive result that administering the traps prevents viral escape even after 15 passages. Finally, the therapeutic potential of evolved ACE2-Fc candidates is explored with a pharmacokinetics study in mice and a 5-day efficacy study in hamsters.

This work was very exciting and thorough. I think it should be published, but would appreciate if the authors can address some points:

1. The authors convincingly show that their ACE2-Fc therapeutic candidates outcompete the wild-type protein in competitive binding assays and a mass photometry experiment, and go on to do additional biochemistry and structural biology to show why this is. However, the design of the main experiment to compare binding among different evolved ACE2s (which results in the data that appear in Extended Data Figs. 1c, 2, 6, and 8f) is a bit sloppy. First of all, it is not a neutralization assay – there are no viruses – it's a competitive binding assay. Secondly and more importantly, the IC50s for the same ACE2 constructs are all different in each figure, presumably because the authors did not use the same concentration of spike protein in the different experiments. In fact they only used set concentrations of spike for the experiment in Extended Data Fig. 6. So there is no way to know how much ligand depletion may be occurring for each experiment and since the data are only shown in singlicate, it reduces the confidence in each result – the same proteins behave differently in different experiments since the spike protein was at unknown concentrations in each one and we cannot compare among the results. It would be great if the authors could repeat the experiment in ED Fig. 1c in triplicate with a defined spike concentration for each ACE2-Fc, show some error bars for this experiment, and clarify the captions for the other figures that involve this type of experiment to explain that the data for each protein are relative to each other because of the spike dilution.

2. It is interesting that there were no affinity-enhancing PD2 mutations from evolution from wild-type ACE2 as well as in the tight-binding PD1 mutants, given that other studies (e.g. KK Chan et al., 2020) found beneficial mutations in this region. Can the authors comment on this result more?

3. It is interesting that 3J113v2 is uniquely unable to bind SARS-CoV-1 RBD – can the authors briefly comment on why this may be, given the mutations in 3J113v2 and the differences in SARS-CoV-1 RBD and SARS-CoV-2 RBD?

4. The study includes a really nice experiment to lock the closed conformation of ACE2 with a disulfide bond to determine how this conformation influences the affinity, and find that it is likely a stability effect. But the enzymatic activity assay that they use depends on a fluorogenic substrate cleavage, and doesn't measure angiotensin II cleavage. It turns out that these two substrates can be cleaved differently by ACE2 mutants (Liu, Xie, Gao and Jin, "Designed variants of ACE2-Fc that decouple anti-SARS-CoV-2 activities from unwanted cardiovascular effects", 2020). I don't think

it's necessary to do the angiotensin II cleavage experiments, but the authors should note in the main text that their experiment does not directly measure that.

5. All of the experiments in Extended Data Fig. 10 are in mice, right? It is surprising that the half-life for ACE2-Fc is so short. Iwanaga et al., 2020, showed that ACE2-Fc has a 6-day-long half-life in mice. Can the authors please clarify whether ED Fig. 10b is also in mouse, and why these ACE2-Fc have a considerably shorter half-life than was found in the previous study?

6. It would also strengthen the paper if the discussion included some analysis on how the affinity-enhancing mutations that were discovered in this study are similar/different from those found in prior work building engineered ACE2-Fc molecules.

Thank you for the opportunity to read about this research.

Response to Reviewers

We thank the reviewers for their interest in our manuscript and constructive suggestions. We have addressed all of the reviewers' concerns point-by-point below, which has significantly improved the manuscript. Importantly we confirmed therapeutic effect of engineered ACE2 at 2 days post-infection, more practical regimen, as well as intact neutralization efficiency against current major SARS-CoV-2 variants. Next, we reevaluated the half-life in polyclonal antibody-based ELISA and found that it was ~25 hr, longer than previous data. We also discussed about the affinity-enhancing mutations found in this study and previous reports and about the concerns that should be resolved in future preclinical study.

REVIEWER COMMENTS

Reviewer #1 (Remarks to the Author):

In this manuscript the authors used human cell-based directed evolution to select for an ACE2 protein with tighter binding to the RBD of the spike protein. After three cycles of random mutagenesis they obtained a 100-fold tighter binding ACE2 variant. Using SARS-CoV-2 they showed that no escape mutant was developed after 15 passages. This point is important, particularly now, that escape mutants take hold. They solved the x-ray structure of the complex, and showed that the high affinity ACE2 is protective to hamster, if given at a dose of 20 mg/Kg, 2 hr after their infection with CoV-2. This is a well performed and nice study on a potential drug against COVID-19.

We thank the reviewer for appreciating the importance of our study and recognizing the potential of our mutant ACE2 as a therapeutic against COVID-19.

The major problem here is with novelty. A number of studies were published, using similar strategies. For example, Wells et al published in PNAS (doi.org/10.1073/pnas.2016093117) a similar study, optimizing ACE2 for spike binding. About the same time, Procko published in science a study of ACE2 optimization for the same purpose (DOI: 10.1126/science.abc0870). Dang published a study on making an optimized ACE2 trimer, with super high neutralization efficacy towards CoV2 (Cell Research volume 31, pages98-100(2021)). In the above cited studies, the advantage of using a decoy receptor, as it avoids virus escape mutants is also discussed. In addition to these, other studies using similar strategies (on the basis of ACE2 optimization) have been deposited in bioRxiv.

Yes, we understand that similar (but distinct) ACE2 mutants have been reported as therapeutic candidates

and that the strategy itself is not novel. However, we believe that the current work contains high degree of novelty and impact from the following reasons. First, we provide crystal structure of the mutant ACE2 in complex with spike receptor-binding domain (RBD), which has not been reported for any of the ACE2 mutants published before. This information not only enabled us to elucidate the mechanism of the affinity improvement by the mutations, but also provided, unexpectedly, the rationale for a further engineering of the protein to suppress its undesirable peptidase activity and to increase the stability. Second, we provide concrete experimental evidence that the receptor decoy strategy is in fact effective in preventing the emergence of resistant viruses. This was shown by the sustained effectiveness of our mutant ACE2 (the decoy agent) in the virus elimination activity over the repeated treatment and re-infection cycles in cultured host cells. Lastly and most importantly, we have shown the therapeutic potential of the engineered ACE2 in animal model. Animal study of SARS-CoV-2 had been difficult because most virus strains cannot infect mouse or rat. We therefore utilized hamster model and found that the agent was capable of greatly reducing the symptoms of hamster infected by the SARS-CoV-2 virus when administered 2-hr after the virus challenge. As to the Dang and colleagues' paper, the ACE2 trimer strategy is certainly of great importance and quite effective in preventing in vitro virus infection. It may even be highly effective in animal study. However, this agent is not likely to be considered as therapeutics in humans due to the concern of the immunogenicity, because it contains the trimerization domain (foldon) derived from T4 phage as the key component.

Specific comments:

1. The author referred to the potential of the spike to bind 3 ACE2 proteins. It would be appropriate here to cite the work Cell Research volume 31, pages98-100(2021), where this has been done (including a structure).

We thank the reviewer for pointing out this important work. According to the suggestion, we cited this paper in the revised manuscript (page 4, line 29). However, the ACE2 protein they used is a preformed trimer (by virtue of the C-terminal fusion of the trimerization domain foldon), making it highly advantageous in occupying three RBDs simultaneously. Therefore, it remained unclear if all the spike RBDs on the surface of a virus can be occupied by monomeric ACE2 proteins, until we definitely show that all RBDs can be saturated by ACE2 by mass photometry analysis (Extended Data Fig. 4). Furthermore, their cryo-EM map shows weak density in the regions where ACE2 would sit (Fig.S7e), which may indicate partial occupancy or high flexibility. Therefore, comparison with the Dang et al paper in fact helped to emphasize the potential of our mutant ACE2 as the soluble virus-neutralizing agent. We thank the reviewer again for this important suggestion.

2. An interesting point to consider is the different mutations discovered in this study, versus other studies where ACE2 was evolved to bind tighter to the RBD. The authors should discuss this issue, as it may allow for even higher binding affinity to be generated.

Computational mutant scanning has indeed showed that some of the differences in the residues mutated in the different studies can be attributed to the non-optimal receptor amino acids at some positions within the ACE2-spike interface.

We thank the reviewer for this valuable suggestion to discuss more on the potential improvement of the affinity by incorporating information reported by others, which we were reluctant in the original submission due to the speculative nature of the subject. According the suggestion, we now have Table S2 so that all the mutated residues found by us, Procko et al, and Wells et al are listed. Mutations are frequently found in the residues located at the interface with spike RBD including A25, K31, H34, E35, and L79, although the type of amino acid replacement were variable and no strong correlation patterns in the occurrence of particular mutations have emerged. Interestingly, mutations found by us and Wells' paper are limited within PD1, whereas Procko's group found mutations both in PD1 and PD2. We already knew that additional mutations in PD2 region to the 3rd generation PD1-limited mutant library failed to isolate better mutants (Extended Data Fig. 1b). Encouraged by the reviewer's suggestion, we tested the combination of our 3N39v2 and individual affinity-enhancing PD2 mutations in Procko's DMS panel. To our surprise, none of them showed increased affinity (newly added Extended Data Fig. 1e). The failure of the cumulative approach suggests that the contribution of each mutation to the increased affinity is highly context-dependent, pointing toward the need for obtaining structural information at atomic resolution for more mutants to instruct designs of even better therapeutic candidates. Another important point to be considered is the way of performing the selection. The SPR data suggest that promising mutants isolated in three studies (i.e., N39, J113, and J320 in ours, v2.4 in Procko, and #313 in Wells) all show very slow dissociation rate from the spike RBD, suggesting that slowing down the dissociation further may not result in the higher FACS signal during the selection cycles, precluding the isolation of clones with even higher affinity. Therefore, selection methods orthogonal to the current ones, such as the computational mutant scanning as the reviewer pointed out, may need to be combined to obtain further ACE2 mutants. The above discussion was included in the discussion section of the revised manuscript (page 9, line 8-27).

New Extended Data Fig. 1e

3. In the animal experiment a very high amount of ACE2-FC was given (20 mg/Kg, IP). I would

be worried that this high amount of mutant ACE2 may result in generation of antibodies against ACE2, which could be harmful for the future for the patient. Was this evaluated? Despite the high dose of ACE2, the therapeutic effect (reduction of viral titer and cytokine secretion) was significant, but smaller than expected. Why not use inhalation of a much lower dose? What would happen if the drug would be given 2 days after virus inoculation, to better mimic treatment in human. Will there still be a therapeutic effect?

We do consider this issue seriously because we are trying to develop anti-COVID-19 drugs based on the discovery described in the current work. To predict potential antigenicity of the mutants, we performed T-cell proliferation assay (outsourced to Proimmune Ltd). [Redacted]

[Redacted]

It is crucial to avoid mutation contained in this stimulative peptide in the final drug candidate to prevent emergence of anti-drug antibody and cross-reactivity against endogenous ACE2. This information is provided for the review purpose only and we cannot include the data in the revised manuscript not only because of the proprietary nature of the content, but also because it is beyond the scientific scope of the current paper.

As to the route of administration, we are very much interested in the inhalation exposure regimen and will consider that option during our preclinical trial study. However, direct delivery into the respiratory

system is generally suited for small molecules. It was reported that nanobodies and small proteins can be effective in this regimen owing to the small size and high heat stability. As our ACE2-Fc is larger than a regular monoclonal antibody and unlikely to show high stability comparable to miniproteins, our current thinking is that the intravenous administration should have the highest chance of success.

As to the efficacy of the ACE2-Fc given at later time point, we agree with the reviewer that it is ideal to see antiviral effect with longer interval to better mimic human treatment condition. We therefore performed additional experiments where the engineered ACE2 was administered to hamster at 2 days post-infection. As shown in the new Extended Data Fig. 14, this treatment showed modest but significant reduction of virus RNA and cytokine expression. It is a promising result for clinical use in human COVID-19, and we are glad that we did this additional experiment. Of course, this is just a small step and it is critical to optimize the administration regimen in more relevant animal models and we will work through it in preclinical study. Thank you very much for giving us this opportunity.

New Extended Data Fig. 14

Reviewer #2 (Remarks to the Author):

The manuscript entitled: "Engineered ACE2 receptor therapy overcomes mutational escape of SARS-CoV-2" by Higuchi et al. describes how sACE2 can be used therapeutic lead to treat COVID-19 patients. The idea for using sACE2 as COVID-19 treatment is not new however, the authors developed improved therapeutic leads with higher affinity and extended half-life.

The research group started with mutating and screening for sACE2 variants that exhibit nM affinities to the SARS-CoV-2 receptor binding domain. The focused-on mutation at the spike protease domains (PD) as it is known to harbor the interface to viral spike protein. The screening was carried out with a reporter utilizing HEK293 cells. High-affinity sACE2 was further improved by identification of essential mutations and the discovered variants were

designated as version 2 of the leads. The high-affinity sACE versions exhibited also very slow off-rate compared to the WT ACE2.

Mass photometry analysis indicated that that the mutants can saturate all three RBDs on the SARS-CoV-2 spike trimer at low concentration, a property highly desirable for the virus-neutralizing agent.

Next, to reduce the fast clearance of the lead, they fused the mutated version of sACE to IgG1 Fc and evaluated their neutralization capacity against the SARS-CoV-2 in TMPRSS2-expressing VeroE6 cells and found that each mutant ACE2-Fc demonstrated a significant neutralizing effect even in 100-fold lower concentration than WT.

By structure analysis, the authors show that mutations collectively lead to the 100-fold increase in the overall affinity characterized by their binding affinities by surface plasmon resonance (SPR). The authors also learned from the structural data that the inhibitor-blocked ACE2 in the closed conformation can still engage SARS-CoV-2 RBD, suggesting that caution must be taken when considering the possibility of using ACE2 inhibitors as the therapeutics against coronavirus infection.

They further confirmed that the engineering sACE-Fc fusion does not generate escape mutants after 15 passages and also confirmed that mutant ACE2-Fcs effectively neutralized H4-resistant F490V mutated virus, as well as escape mutants from COVID-19 convalescent plasma, Δ 69-70/D796H6, and insertion of N-linked glycan sequence.

Finally, they selected the 3N39v2-Fc mutant based on its high expression efficiency to evaluate the therapeutic potential in a hamster model of COVID-19 characterized by rapid weight loss and severe lung pathology. They show that hamsters that received control-Fc lost 4.3% of body weight, whereas those treated with 3N39v2-Fc gained 7.3% similarly to the unchallenged control 5 days post-infection. They also analyzed the content of SARS-CoV-2 in lungs as functional virus particle and genome RNA copy number, and it was revealed that both were significantly decreased in 3N39v2-Fc-treated group.

Overall, the methods and experimental setups are solid and the data is convincing. The authors utilized several orthogonal approaches to well-characterized the discovered biological lead. However, there are two main concerns regarding the potential of such biologics to treat COVID-19 patients in the “real world” .

First, what will be the effective dosage to be given to COVID-19 patients? given there is

inter-patient variability of viral loads, what will be the effective dose to be used? and what will be the administration regimen, i.e. how many times will a patient need to be treated with the engineered sACE2-Fc so it will show high efficacy?

We thank the reviewer for the careful reading and accurately understanding the content of the paper in great detail. It is natural that experts would be interested in the detail of the administration regimen since the ultimate goal is to develop effective therapeutics to fight against COVID-19. We are aware that it is essential for us to optimize the administration regimen as early as possible. Since our engineered ACE2-Fc has relatively short half-life in blood (~25 hours), we think that, if we are to use the current version of ACE2-Fc, a daily administration is necessary to obtain full treatment effectiveness. Although we will work through regimen optimizations in preclinical study, we think that thorough investigation on the detailed therapeutic strategy is beyond the scope of the current paper, which is aimed at reporting a promising drug lead with a clear mechanism of action.

Derived from the first concern, will the treatment strategy (i.e. dosage and a number of administrations) elicit an immune response in the recipient. Some biologics are notorious for their immunogenicity and inducing anti-drug antibodies. This may occur even with monoclonal antibodies that are fully human. In the case of the engineered sACE2 and with dependence on the dosage strategy, there is a possibility that the drug will mount an anti-drug immune response by breaching tolerance (as it was mutated and cannot be considered WT). In such an event, there is a danger that the patient will be generated immune memory to the self-ACE2 that can in-turn induce long-term adverse effects. It will be good if the authors address both of these concerns.

We do consider this issue seriously because we are trying to develop anti-COVID-19 drugs based on the discovery described in the current work. To predict potential antigenicity of the mutants, we performed T-cell proliferation assay (outsourced to Proimmune Ltd). [Redacted]

[Redacted]

So we learned that we should avoid mutation contained in this stimulative peptide in the final drug candidate to prevent emergence of anti-drug antibody and cross-reactivity against endogenous ACE2 as pointed out by the reviewer. This information is provided for the review purpose only and we cannot include the data in the revised manuscript not only because of the proprietary nature of the content, but also because it is beyond the scientific scope of the current paper. Nevertheless, we added a paragraph describing the importance of considering the potential immunogenicity and the potential routes to minimize the risk at the end of the discussion section (page 10, line 5-11).

Reviewer #3 (Remarks to the Author):

The Authors have provide evidence that engineered ACE2 have therapeutic potential in hamster animal model beyond the escape mutant SARS-CoV-2.

We thank the reviewer for appreciating the importance of our study and recognizing the potential of our mutant ACE2 as therapeutics against COVID-19.

Major comments:

* The Authors need to include virus of concerns (VOC) such as 501y.v1, 5017.v2 and p1 in their animal model.

We totally agree that the therapeutic candidates should be tested against multiple current major SARS-CoV-2 mutants virus strains, ideally those which cause serious concern in the society. These mutants are reported to have higher transmissibility due to N501Y, and two of them (501y.v2 (SA-B.1.351) and BR-P.1) have additional mutation E484K that render them highly resistant against convalescence and vaccine. In response to this comment, we first examined the neutralization capacity of our mutant ACE2-Fcs against these three variants in cultured cells. We did succeed in showing the effectiveness of engineered ACE2 against these VOCs in both pseudotyped and authentic virus formats, which are now included as new fig 4 in the revised manuscript. We then went on to evaluate the therapeutic potential of 3N39v2 against 501y.v1 (UK-B.1.1.7) and BR-P.1 in hamsters. To our surprise, however, it failed to reduce virus copy number in the infected lungs. As the macroscopic lung pathology of mice received these VOCs were much severer than that caused by Wuhan strain, these variants seem to have stronger virulence in hamsters, which may be the reason why our ACE2-Fc could not show clear protection. Whatever the reason is, we cannot include data for animal protection using VOC as the reviewer recommended. However, we believe that the in vitro neutralization data warrant that our mutant ACE2 would have sufficient therapeutic potential against VOC.

New Fig.4

* The Author have used VeroE6 cells to push for escape mutant SARS-CoV2. The Authors need also include other cell lines and human primary cell lines to study the escape mutant.

We agree that experimental data obtained with a variety of cells, particularly human cell lines, may make the evidence stronger. However, we think that the use of Vero6 cells in the experiment of escape virus emergence can be well justified by the following reasons. First, virus genome mutation is generated by low fidelity RNA-dependent RNA polymerase coded in viral genome, not by host-coded proteins, so the mutational events are minimally affected by the species of host cells. Furthermore, escaping character toward infection-neutralizing drugs is determined by the change in the affinity between drugs and virus receptor, ACE2. Fortunately, the homology of ACE2 between human and green monkey (*Cercopithecus aethiopsis*) is quite high (94.67%) and most importantly, it is 100% conserved for the interface residues to spike RBD (Scientific Reports, 2021 11: 1702, PMID: 33462320; see figure below). Therefore, the results of Vero 6 cell experiments can be applied to the escaping character in human. We also want to point out that the virus growth is generally very slow in human cells and it will take months to confirm the results if we are to use human cells.

Species	Host coronavirus	ACE2 binding hotspots similarities with hACE2															Total similarity
		S19	Q24	D30	K31	H34	E35	E37	D38	Y41	Q42	L79	M82	Y83	K353	R393	
Homo sapiens	SARS-CoV-2, SARS-CoV, MERS	S19	Q24	D30	K31	H34	E35	E37	D38	Y41	Q42	L79	M82	Y83	K353	R393	15/15
Macaca mulatta	n/a	S19	Q24	D30	K31	H34	E35	E37	D38	Y41	Q42	L79	M82	Y83	K353	R393	15/15
Macaca fascicularis	n/a	S19	Q24	D30	K31	H34	E35	E37	D38	Y41	Q42	L79	M82	Y83	K353	R393	15/15
Cercopithecus aethiops	n/a	S19	Q24	D30	K31	H34	E35	E37	D38	Y41	Q42	L79	M82	Y83	K353	R393	15/15
Mustela putorius furo	n/a	S19	L24	E30	K31	Y34	E35	E37	E38	Y41	Q42	H79	T82	Y83	K353	R393	9/15
Canis lupus familiaris	Canine CRCoV	S19	L23	E29	K30	Y33	E34	E36	E37	Y40	Q41	L78	T81	Y82	K352	R392	10/15

* the Authors should include more Cytokines in figure 4.

According to this comment, we performed additional experiments to evaluate quantity of more cytokines. As a result, we found that proinflammatory IFN- γ , IFN- λ , and inflammation-related IL-10, TGF- β were also significantly attenuated in 3N39v2-Fc-treated hamsters. We thank the reviewer for the useful suggestion to make our data more convincing. These results were included as new Extended Data Fig. 13.

New Extended Data Fig. 13.

* It is of interest to analyse the therapeutic effect of ACE2 at different time post infection. It is not practical to treat patient just after infection. the antiviral need to be used after symptoms appear.

We agree with the reviewer that it is ideal to see antiviral effect with longer interval to better mimic human treatment condition. We therefore performed additional experiments where the engineered ACE2 was administered to hamster at 2 days post-infection. As shown in the new Extended Data Fig. 14, this treatment showed modest but significant reduction of virus RNA and cytokine expression. It is a promising result for clinical use in human COVID-19, and we are glad that we did this additional experiment. Of course, this is just the first step and it is critical to optimize the administration regimen in more relevant animal models and we will work through it in preclinical study. Thank you very much for giving us this opportunity.

New Extended Data Fig. 14

Reviewer #4 (Remarks to the Author):

Review for

“Engineered ACE2 receptor therapy overcomes mutational escape of SARS-CoV-2”

Higuchi, Suzuki, Arimori, Ikemura et al.

In this nice study, the authors establish a mammalian display platform for directed evolution of ACE2 to build ACE2-Fc receptor decoys as therapeutic candidates to treat SARS-CoV-2 infections. They demonstrate neutralization efficacy of several of their candidates in pseudoviral and authentic SARS-CoV-2 viral neutralization cell culture assays. The mechanistic basis for increased affinity for spike binding in the evolved ACE2 candidate with the lowest IC₅₀ is revealed by an X-ray crystal structure of the complex. The authors perform an experiment to compare the mutational escape of SARS-CoV-2 when treated with a neutralizing antibody to their receptor trap, which gives the impressive result that administering the traps prevents viral escape even after 15 passages. Finally, the therapeutic potential of evolved ACE2-Fc candidates is explored with a pharmacokinetics study in mice and a 5-day efficacy study in hamsters.

This work was very exciting and thorough. I think it should be published, but would appreciate if the authors can address some points:

We thank the reviewer very much for highly valuing our work and expressing strong support for the potential publication of this paper.

1. The authors convincingly show that their ACE2-Fc therapeutic candidates outcompete the wild-type protein in competitive binding assays and a mass photometry experiment, and go on to do additional biochemistry and structural biology to show why this is. However, the design of the main experiment to compare binding among different evolved ACE2s (which results in the data that appear in Extended Data Figs. 1c, 2, 6, and 8f) is a bit sloppy.

First of all, it is not a neutralization assay - there are no viruses - it's a competitive binding assay. Secondly and more importantly, the IC50s for the same ACE2 constructs are all different in each figure, presumably because the authors did not use the same concentration of spike protein in the different experiments. In fact they only used set concentrations of spike for the experiment in Extended Data Fig. 6. So there is no way to know how much ligand depletion may be occurring for each experiment and since the data are only shown in singlicate, it reduces the confidence in each result - the same proteins behave differently in different experiments since the spike protein was at unknown concentrations in each one and we cannot compare among the results. It would be great if the authors could repeat the experiment in ED Fig. 1c in triplicate with a defined spike concentration for each ACE2-Fc, show some error bars for this experiment, and clarify the captions for the other figures that involve this type of experiment to explain that the data for each protein are relative to each other because of the spike dilution.

We apologize that the way data were presented and the description about the competition assay lacked sufficient accuracy. Yes, these experiments simply measured competition with WT ACE2 in the binding of RBD and cannot be described as neutralization assays. We amended all the misstatements regarding this point (e.g., figure legend of ED Fig 1, 2, 9, page 9-line 5). The reviewer is absolutely right in that a comparison of IC50 values among different datasets is meaningless because the concentrations of RBD and spike trimer could not be set constant. However, we do not intend to compare IC50 values derived from different competition assays. In these experiments, the main purpose was to determine differences in relative competitive strength among mutants included in one dataset, e.g., compare 5 mutants vs WT (ED Fig.1c), compare 7 back-mutated proteins with 3N39 (ED Fig.2b), compare disulfide-stabilized versions with the non-disulfide bonded original proteins (ED Fig.8f), etc. Although the exact concentration of RBD or spike protein contained in the assay tubes are not known except for ED Fig.6, it is set constant within each experiment (because they were assayed at the same time with the same condition), and we believe we can safely discuss the rank order or qualitative differences among mutants contained in the same assay (= data within a single figure panel). We carefully checked that we only compare mutants contained in the same panel in the text. It is true that the data were shown mostly in singlicate. According to the reviewer's comment, we repeated the experiment in ED Fig. 1c with 50-fold dilution of RBD-sfGFP in triplicate. We believe that data accumulation made the evidence more convincing.

2. It is interesting that there were no affinity-enhancing PD2 mutations from evolution from wild-type ACE2 as well as in the tight-binding PD1 mutants, given that other studies (e.g. KK Chan et al., 2020) found beneficial mutations in this region. Can the authors comment on this result more?

This is indeed an interesting and important point. In order to make it easy to compare among high affinity mutants reported by three groups, we now present Table S2. As the reviewer pointed out, Procko and colleagues (KK Chan 2020) found beneficial mutations in PD2 region. Interestingly, mutants isolated by Wells and colleagues did not have mutations in PD2 as in our case (they have mutated H345 etc. but that was to destroy the enzymatic active site). So, we decided to perform additional experiment by combining the 3N39v2 mutant with potentially affinity-enhancing PD2 mutations present in Procko's deep mutagenesis panel including N330Y and A386L, but none of them showed further increase in the competition activity (new Extended Data Fig. 1e). We suspect that these PD2 mutations may have played auxiliary roles when combined with particular PD1 mutation set, but more realistic discussion about the mechanism of affinity enhancement by individual mutations should await structure determination of these mutants. Another important point to be considered is the way of performing the selection. The SPR data suggest that promising mutants isolated in three studies (i.e., N39, J113, and J320 in ours, v2.4 in Procko, and #313 in Wells) all show very slow dissociation rate from the spike RBD, suggesting that slowing down the dissociation further may not result in the higher FACS signal during the selection cycles, precluding the isolation of clones with even higher affinity. Therefore, selection methods incorporating, for example, longer or harsher washing conditions focusing on kinetic off rates may be needed to identify higher affinity ACE2 mutants. The above discussion was included in the discussion section of the revised manuscript (page 9, line 8-27). We thank the reviewer for enlightening us by this comment.

New Extended Data Fig. 1e

3. It is interesting that 3J113v2 is uniquely unable to bind SARS-CoV-1 RBD - can the authors briefly comment on why this may be, given the mutations in 3J113v2 and the differences in SARS-CoV-1 RBD and SARS-CoV-2 RBD?

We thank the reviewer to comment on this point. Structurally, K31M/E35K in 3J113 would act similarly to the K31N/E35K in 3N39, freeing the intramolecular salt bridge and reorienting the K35 toward Q493 (SARS-CoV-2) or N479 (SARS-CoV-1). However, the larger M31 sidechain may not be favored when binding to SARS-CoV-1 RBD, where the abutting L455 is changed to much larger Y442, while small N31 in 3N39 can be accommodated and may even make favorable hydrogen bond with Y442 (Extended Data Fig. 8). Additional figure was created to explain them as Extended Data Fig. 8 and mentioned in the results section.

4. The study includes a really nice experiment to lock the closed conformation of ACE2 with a

disulfide bond to determine how this conformation influences the affinity, and find that it is likely a stability effect. But the enzymatic activity assay that they use depends on a fluorogenic substrate cleavage, and doesn't measure angiotensin II cleavage. It turns out that these two substrates can be cleaved differently by ACE2 mutants (Liu, Xie, Gao and Jin, "Designed variants of ACE2-Fc that decouple anti-SARS-CoV-2 activities from unwanted cardiovascular effects", 2020). I don't think it's necessary to do the angiotensin II cleavage experiments, but the authors should note in the main text that their experiment does not directly measure that.

We agree that this is an important point. We used commercially available ACE2 assay kit that determines MCA fluorescence liberated from the surrogate peptide and didn't directly confirm the cleavage of angiotensin II itself. This point is now noted in the results section.

5. All of the experiments in Extended Data Fig. 10 are in mice, right? It is surprising that the half-life for ACE2-Fc is so short. Iwanaga et al., 2020, showed that ACE2-Fc has a 6-day-long half-life in mice. Can the authors please clarify whether ED Fig. 10b is also in mouse, and why these ACE2-Fc have a considerably shorter half-life than was found in the previous study?

We thank the reviewer for this insightful comment. Yes, the experiments were carried out in mouse. We were also confused about the results inconsistent with some of the reported values, so revisited this experiment. First, we developed our own in-house sandwich ELISA system suitable for the quantification of mutant ACE2-Fc proteins. To this end, we first immunized rabbits with WT ACE2 monomer and obtained high-titer antisera against human ACE2. Then we coupled 3N39 ACE2 onto Sepharose beads, and affinity-purified polyclonal antibody reactive with this mutant from the sera. This ACE2-specific (and also mutant ACE2-crossreactive) IgG was directly labeled with HRP to prepare "detector" reagent, while commercial anti-human IgG1 Fc-specific antibody was used as the "capture" agent. By using this assay system, we could accurately measure the amount of ACE2-Fc proteins in mouse plasma, and in fact found that the half-life was much longer than the previous experiments. As the proteins were administered intraperitoneally, rather than intravenously as in Iwanaga et al paper, the pharmacokinetics could not be interpreted based on a simple one-phase decay. Assuming the two-phase PK model (i.e., the first phase representing the transfer of the agents from the peritoneal cavity to blood and the second phase representing the decay in the blood) and that the first phase would be completed after 24h, the day1-7 data were applied to one phase model, leading to the estimated plasma half-life of about 30hr (ACE2-Fc) and 25 (3N39v2-Fc). These values are much longer than the ones obtained with the commercial ACE2 ELISA kit previously, which may not be accurate enough to quantitate mutant ACE2-Fc, but still much shorter than what Iwanaga et al., 2020 reports. We then realized that Iwanaga et al used LALA mutated Fc to achieve long half-life. Moreover, their sandwich ELISA used Fc antibodies as both capture and

detector reagents, which would detect Fc-only proteins that may be produced during the decay, leading to a potential overestimation of the half-life. LALA mutation that has low affinity to FcγRs is advantageous in PK and may be suitable for the prophylactic purpose. In contrast, our engineered ACE2 employed intact Fc that can bind to FcγRs to activate CD8+ T cell immunity to warrant therapeutic effects in COVID-19 patients. (Emma et al., Cell 2021, Bournazos et al., Nature 2020). Now these results are included along with the new ED Fig. 12, and discussed in more detail in the revised manuscript (page 9, line 28 – page 10, line 4).

New Extended Data Fig. 12.

6. It would also strengthen the paper if the discussion included some analysis on how the affinity-enhancing mutations that were discovered in this study are similar/different from those found in prior work building engineered ACE2-Fc molecules.

This is the point also raised by the reviewer #1. According the suggestion, we now have new table S2 so that all the mutated residues found by us, Procko et al, and Wells et al can be easily compared. As already mentioned in the answer to your 2nd comment and to the reviewer #1's comment, each mutation set is highly context-dependent and it seems difficult to understand the mechanism of affinity-enhancement in each case without 3D structures. We nevertheless included our thoughts regarding this point in the discussion section.

Thank you for the opportunity to read about this research.

No, WE thank this reviewer for the high-standard and very insightful comments and suggestions!

REVIEWERS' COMMENTS

Reviewer #1 (Remarks to the Author):

The author answered by concerns and performed the required experiments. I am still somewhat skeptic concerning the medical use, however, the resistance against upcoming COVID19 variants is impressive. It is also nice to see that the inhibitor works against the already emerging variants. I am still somewhat perplexed on why there is hardly any overlap in the mutations on ACE2 that give higher affinity found in the different studies. I encountered another publication on bioRxiv, doi.org/10.1101/2020.08.12.247940, where a higher affinity binding ACE2 was selected using computational and experimental methods, and again there was no overlap in the specific mutations (although, the positions partially overlap). This fits the view that binding sites are plastic, and that many solutions exist to obtain high binding affinity.

Specific comments:

I am somewhat worried whether the results shown in Fig S1e are not a result of the molar amount of sACE2-sfGFP being limited relative to the molar amount of RBD-sfGFP at high dilutions of the sACE2-sfGFP. As the authors don't provide the molar concentrations but only the dilutions one cannot know. If indeed the molar concentration of sACE2-sfGFP is limiting, the titration curves of all high affinity binders will be the same, not due to the same affinity but to limit in the number of inhibiting molecules. Similar problems can occur in the pseudo virus and virus assays (the molar amount of sACE2 for the high affinity mutants is not sufficient to cover all virus RBDs). As the authors know how much virus was used, and can estimate the number of spike per virus, they can do the simple math to see if there is any problem as such.

Reviewer #2 (Remarks to the Author):

I have read carefully the point-by-point replies by the authors and all of my concerns were addressed.

I hope that further pre-clinical experiments will result in promising results progressing this therapeutic solution towards clinical trials.

This is an important report that is suitable for Nat. Comm. readers.

Yariv Wine

Reviewer #3 (Remarks to the Author):

The Authors have addressed my concern

Reviewer #4 (Remarks to the Author):

I'm satisfied by the authors' response and revisions. Great study - publish it!!

Response to Reviewers

REVIEWER COMMENTS

Reviewer #1 (Remarks to the Author):

The author answered by concerns and performed the required experiments. I am still somewhat skeptic concerning the medical use, however, the resistance against upcoming COVID19 variants is impressive. It is also nice to see that the inhibitor works against the already emerging variants. I am still somewhat perplexed on why there is hardly any overlap in the mutations on ACE2 that give higher affinity found in the different studies. I encountered another publication on bioRxiv, doi.org/10.1101/2020.08.12.247940, where a higher affinity binding ACE2 was selected using computational and experimental methods, and again their was no overlap in the specific mutations (although, the positions partially overlap). This fits the view that binding sites are plastic, and that many solutions exist to obtain high binding affinity.

Thank you for introducing another paper of modified ACE2. This paper computationally identified favorable amino acid mutation and put them all in one mutant. KD value of the mutant is >100 fold better than that of wild type. However, IC50 is not great, as compared with ours or others. So far, the computational design is not perfect. Wells also computationally designed the mutant but additionally performed directed evolution to achieve enough affinity. Baker designed a minibinder (DOI: 10.1126/science.abd9909). His strategy was also combination of computational design and intensive experimental screenings. As pointed, each study found the different high affinity mutants and the question is whether they are one of the best or there are other mutants stronger than them. To discuss about it, it is important to accumulate structural information at atomic resolution as we did.

Specific comments:

I am somewhat worried whether the results shown in Fig S1e are not a result of the molar amount of sACE2-sfGFP being limited relative to the molar amount of RBD-sfGFP at high dilutions of the sACE2-sfGFP. As the authors dont provide the molar concentrations but only the dilutions one cannot know. If indeed the molar concentration of sACE2-sfGFP is limiting, the titration curves of all high affinity binders will be the same, not due to the same affinity but to limit in the number of inhibiting molecules. Similar problems can occur in the pseudo virus and virus assays (the mol amount of sACE2 for the high affinity mutants is not sufficient to cover all virus RBDs). As the authors know how much virus was used, and can estimate the number of spike per virus, they can do the simple math to see if there is any problem as such.

At first, we also had the same concern that this assay had the limit in the sensitivity to find better candidates. One simple answer is the pseudovirus or authentic virus experiments in figure 4. Monoclonal antibody REGN10933 neutralized virus at ~10-fold lower concentration, as compared with mutant ACE2-Fc. And, in the similar competitive binding assay with defined molar ACE2 and spike trimer in supplementary fig. 6, we additionally performed the assay with different concentration of ACE2-Fc and observed that ~105nM ACE2 failed to completely block 30nM S-trimer (90nM S monomer). These results indicates that incomplete inhibition is not due to the limit in the number of inhibiting ACE2 and there is the window to detect higher affinity mutants.